# Optimize Any Topology: A Foundation Model for Shape- and Resolution-Free Structural Topology Optimization

**Amin Heyrani Nobari**
Massachusetts Institute of Technology
Cambridge, MA, 02139
`ahnobari@mit.edu`

**Lyle Regenwetter**
Massachusetts Institute of Technology
Cambridge, MA, 02139
`regenwet@mit.edu`

**Cyril Picard**
Massachusetts Institute of Technology
Cambridge, MA, 02139
`cyrilp@mit.edu`

**Ligong Han**
Red Hat AI, MIT-IBM Watson AI Lab
Cambridge, MA, 02139
`ligong.han@ibm.com`

**Faez Ahmed**
Massachusetts Institute of Technology
Cambridge, MA, 02139
`faez@mit.edu`

## Abstract

Structural topology optimization (TO) is central to engineering design but remains computationally intensive due to complex physics and hard constraints. Existing deep-learning methods are limited to fixed square grids, a few hand-coded boundary conditions, and post-hoc optimization, preventing general deployment. We introduce Optimize Any Topology (OAT), a foundation-model framework that directly predicts minimum-compliance layouts for arbitrary aspect ratios, resolutions, volume fractions, loads, and fixtures. OAT combines a resolution- and shape-agnostic autoencoder with an implicit neural-field decoder and a conditional latent-diffusion model trained on OpenTO, a new corpus of 2.2 million optimized structures covering 2 million unique boundary-condition configurations. On four public benchmarks and two challenging unseen tests, OAT lowers mean compliance up to 90% relative to the best prior models and delivers sub-1 second inference on a single GPU across resolutions from 64 × 64 to 256 x 256 and aspect ratios as high as 10:1. These results establish OAT as a general, fast, and resolution-free framework for physics-aware topology optimization and provide a large-scale dataset to spur further research in generative modeling for inverse design. Code & data can be found at https://github.com/ahnobari/OptimizeAnyTopology.

## 1 Introduction

Foundation models—large models trained on broad data that can be adapted to many downstream tasks—have transformed modern AI. They underpin advances in vision, language, and multimodal reasoning, powering systems such as large language models (LLMs) and image generators that now drive progress across domains, including scientific breakthroughs in protein folding [21] and drug discovery [16]. In contrast, engineering design remains dominated by *inverse problems*, where the goal is to infer a design that *meets stated constraints while maximizing performance objectives*. Designing a bridge, for example, may require allocating a fixed mass of steel so the finished span withstands specified loads with required stiffness. Because such performance maps are highly nonlinear, data-intensive, and constraint-sensitive, the field continues to rely on direct optimization [6] or narrowly scoped deep-learning surrogates and generative models [38]. Limited datasets and the need for high *precision* have so far hindered the emergence of foundation models for engineering design [38].

39th Conference on Neural Information Processing Systems (NeurIPS 2025).

Structural topology optimization (TO) exemplifies these challenges [49] and presents a good testbed for developing engineering foundation models. TO seeks the optimal material distribution within a design domain to maximize physics-based performance metrics such as stiffness or strength. Solving TO conventionally involves repeated Finite Element Analysis (FEA) and gradient-based updates until convergence [3, 41, 50]. This process is computationally expensive and must be entirely restarted when boundary conditions, geometry, or load cases change, limiting its practicality in interactive or large-scale design settings. Solving the underlying PDEs on arbitrary shapes and boundary conditions also yields a huge, non-convex search space, where tiny design changes can trigger failure; high *precision* is therefore essential.

To alleviate these costs, deep learning methods have emerged to accelerate or replace optimization [38, 49]. While comprehensive reviews detail this expansive field [49, 62], our focus is on the evolution of models that *directly* predict near-optimal topologies from problem specifications (boundary conditions, forces, volume fraction) [13, 14, 20, 30, 33–35]. By directly predicting optimal topologies, these aim to substitute the entire optimization process with a rapid predictive framework. These typically employ Deep Generative Models (DGMs)—such as Variational Autoencoders (VAEs), Generative Adversarial Networks (GANs) [2, 26, 32, 37, 48, 61, 63], and more recently Denoising Diffusion Probabilistic Models (DDPMs) [13, 14, 30]—to frame TO as a conditional generation task. Despite numerous innovations, these direct prediction approaches have faced significant criticism by a few TO researchers for their limitations [34, 35, 62].

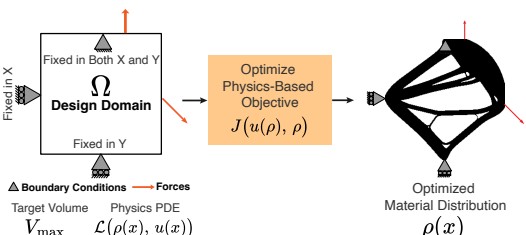

**Figure 1:** In TO, the objective is to distribute material in a domain (a density field $\rho(x)$) to obtain optimal physics-based performance. Above, we show an example of TO for maximum stiffness, given boundary conditions of material supports and forces applied.

The most significant issue, termed the generalizability challenge, is characterized by the inability of current models to manage arbitrary boundary conditions and forces [62]. Current direct prediction methods are trained on finite, often small, sets of predefined problem configurations (e.g., a maximum of 210 unique setups [35]). Consequently, they fail to generalize to unseen or more complex settings, fundamentally limiting their practical utility. Key limitations include: 1) Restricted domain geometry, often confined to fixed square grids [13, 14, 30, 33], rendering many models limited proofs-of-concept. 2) Simplistic problem representations, frequently adapting FEA fields as images for standard vision models (e.g., CNNs) [34]. While recent works [34, 64] show promise in addressing some of these domain and representation issues, a more critical limitation persists: 3) Poor generalizability. Existing models are trained on narrow, controlled problem sets (e.g., only 42 boundary condition configurations [14, 30, 34]) and fail on unseen conditions, considered a major hurdle by critics [62].

We introduce the **Optimize Any Topology (OAT)** framework, a pre-trained latent diffusion model for *general* structural TO focused on minimum compliance. OAT represents the first attempt to simultaneously address the representation and shape/resolution limitations noted in prior work, while also tackling the generalizability challenge. It is positioned as a foundational step towards TO models capable of handling *any* arbitrary boundary conditions and domain shape/resolution. Our foundational scope refers specifically to the model's ability to generalize across arbitrary boundary conditions, forces, and resolutions—capabilities that have not been demonstrated in prior topology optimization research. We emphasize that OAT is focused on the minimum-compliance problem under linear elasticity and does not encompass all forms of physics-driven topology optimization. Thus, rather than claiming a universal foundation model for physics-based design, we present OAT as the first step toward such general frameworks: a foundation model that captures the minimum compliance problem in its most general structural and geometric form.

The primary contributions of this work are as follows: 1) OpenTO Dataset: We introduce OpenTO, the largest and first general-purpose topology optimization dataset, comprising 2.2 million optimized structures with fully randomized boundary conditions, loads, shapes, and resolutions. OpenTO establishes a scalable foundation for data-driven research in physics-based structural design. 2) OAT Framework: We propose Optimize Any Topology (OAT), the first deep learning framework capable of addressing the general topology optimization problem—handling arbitrary boundary conditions and resolutions within a single model. 3) State-of-the-Art Performance: Through extensive experiments, we demonstrate that OAT achieves up to 10x lower compliance error than prior models on established benchmarks, outperforming all existing architectures even without post-optimization refinement. It also performs better than existing methods on the foundational dataset of OpenTO. 4) Scalable Inference: We show that OAT achieves sub-second inference on a single GPU and scales efficiently to higher resolutions, maintaining nearly constant inference time from 64 x 64 to 256 x 256

grids—where prior methods experience severe slowdowns—thus enabling near real-time topology generation across diverse design problems.

## 2 Preliminaries

This section introduces the preliminaries of topology optimization and then examines the challenges and criticisms leveled towards deep learning approaches for TO.

### 2.1 Topology Optimization

Structural Topology Optimization (TO) is the process of determining the optimal distribution of a limited amount of material within a defined design space to maximize (or minimize) specific physics-based performance metrics. Let $\Omega \subset \mathbb{R}^d$ represent a bounded design domain with boundary $\Gamma = \partial\Omega$. The system's physical behavior is described by a state field $u : \Omega \to \mathbb{R}^m$ (e.g., displacement), which is the solution to a governing partial differential equation (PDE). For simplicity, considering only Dirichlet boundary conditions, this PDE takes the form:

$$\mathcal{L}\big(\rho(x),\, u(x)\big) = f(x) \quad \text{in } \Omega, \qquad u(x) = g(x) \quad \text{on } \mathcal{D} \subset \overline{\Omega} \tag{1}$$

Here, $\rho(x) : \Omega \to [0, 1]$ is the material density at each point $x$, acting as the design variable we seek to optimize. $\mathcal{L}$ is a differential operator representing the underlying physics (e.g., elasticity), $f(x)$ represents applied forces or sources, and $g(x)$ prescribes values for $u(x)$ on a portion of the domain $\mathcal{D}$ (which can be on the boundary $\Gamma$ or within $\Omega$). The general continuous TO problem is to find the material density distribution $\rho(x)$ that minimizes a performance objective $J\big(u(\rho),\, \rho\big)$, such as structural compliance (inverse of stiffness) or thermal resistance. This is subject to the governing PDE (Eq. 6) and a constraint on the total material volume $V_{\max}$:

$$\min_{\rho(\cdot)} \; J\big(u(\rho),\, \rho\big) \quad \text{s.t.} \quad \mathcal{L}\big(\rho, u\big) = f(x),\; u(x) = g(x) \quad \text{on } \mathcal{D}\,, \int_{\Omega} \rho \, \mathrm{d}x \leq V_{\max}. \tag{2}$$

This process and the overall problem are depicted in Figure 1. Analytical solutions to the PDE in Eq. 6 are generally unobtainable for complex geometries and material distributions. These practical difficulties necessitate a numerical approach to TO. This is usually done by discretizing the domain into a mesh and solving the problem using Finite Element Analysis (FEA). In this paper, we focus on a common type of topology optimization problem, the minimum compliance TO. The goal in minimum compliance TO is to distribute material to minimize compliance (i.e., maximize stiffness) given some boundary conditions and forces (Figure 1). We detail the underlying physics and optimization scheme for minimum compliance in Appendix A. Despite the cost and complexities of conventional optimization, optimizers such as SIMP [3, 41] are generalizable, meaning that the domain $\Omega$ can have any arbitrary shape, loads may be applied at any node of the FEA mesh and in any direction/magnitude, and the boundary conditions may also be applied in any location and for any direction. As such, a general framework should, in theory, be able to handle these generalities. This, however, proves to be rather difficult in deep learning frameworks built for TO.

### 2.2 Diffusion Models

Denoising Diffusion Probabilistic Models (DDPMs) [18] are a class of generative models that learn to synthesize data by reversing a noise corruption process. Given a data sample $x_0$ from a distribution $q(x_0)$, a fixed forward diffusion process gradually adds Gaussian noise over $T$ discrete time steps. This process can be characterized by $x_t = \sqrt{\bar{\alpha}_t}x_0 + \sqrt{1 - \bar{\alpha}_t}\epsilon$, where $\epsilon \sim \mathcal{N}(0, I)$ is random noise, and $\bar{\alpha}_t$ is a predefined noise schedule that decreases from $\bar{\alpha}_0 \approx 1$ to $\bar{\alpha}_T \approx 0$, such that $x_T$ is approximately distributed as $\mathcal{N}(0, I)$. Typically, a neural network, $\epsilon_\theta(x_t, t)$, is trained to reverse this process by predicting the noise component $\epsilon$ added at time step $t$ to the noisy sample $x_t$. The standard training objective, derived from a variational lower bound on the data log-likelihood, simplifies to minimizing the mean squared error between the true and predicted noise:

$$\mathcal{L}_{\text{DDPM}} = \mathbb{E}_{x_0, t, \epsilon}\left[\|\epsilon - \epsilon_\theta(x_t, t)\|^2\right]. \tag{3}$$

Once trained, samples are generated by iteratively applying the learned reverse transition $p_\theta(x_{t-1}|x_t)$, starting from $x_T \sim \mathcal{N}(0, I)$. Alternative parameterizations for the reverse process exist. For instance, the model can be trained to predict a "velocity" term $v_\theta(x_t, t)$, such as $v = \sqrt{\bar{\alpha}_t}\epsilon - \sqrt{1 - \bar{\alpha}_t}x_0$, which can offer benefits in training stability, uniform signal to noise ratio and sample quality [22, 44]. The objective then becomes $\mathcal{L}^{(v)} = \mathbb{E}_{x_0, t, \epsilon}\left[\|v_\theta(x_t, t) - v\|^2\right]$.

For more efficient sampling, Denoising Diffusion Implicit Models (DDIMs) [53] introduce a non-Markovian reverse process that allows for deterministic generation and significantly fewer sampling steps. DDIMs achieve this by deriving an update rule that directly predicts an estimate of $x_0$ (or uses $\epsilon_\theta$) to take larger, deterministic steps towards the clean data sample. In this way, DDIMs enable fewer sampling steps while retaining quality – crucial in the TO context, where the primary benefit of deep learning is solution speed. In our work, we use the "velocity" term formulation and DDIM rather than the vanilla DDPM objective.

## 3   Related Works

We now discuss prior works closely related to our framework, building upon the established TO background and our paper's motivation.

**Diffusion Models and Resolution Free Variants.**   Originating with Sohl-Dickstein et al. [52], diffusion models [54–56] are now state-of-the-art for conditional and unconditional image synthesis [10, 19, 31, 42, 43], offering stable training over GANs [15] and utility for inverse problems [57], an important feature for TO. Their iterative nature means slow inference, a drawback partially addressed by faster sampling [24, 29, 45]. Nevertheless, pixel-space diffusion remains intensive for high resolutions, limiting prior TO works [34]; latent space compression offers a path to mitigate these issues. Our work aligns with Latent Diffusion Models (LDMs) [40], which perform diffusion in a compressed latent space typically learned via autoencoders [51, 59]. Critical to this paradigm is the nature of the latent representation upon which the diffusion model operates. Most conventional autoencoders work on fixed-resolution/shape images/domains, unsuitable for TO's need for arbitrary aspect ratios and resolutions, thus motivating resolution-free latent representations. Recent efforts extend diffusion to arbitrary resolutions [5, 8, 12, 23, 27]. One such approach performs diffusion in infinite-dimensional continuous spaces using neural operators [5, 12, 23, 25, 27], offering training scalability but slow high-resolution inference as all cells/pixels must be sampled in inference. Alternatively, resolution-free autoencoders with finite-dimensional latents [8] preserve LDM efficiency, often using Implicit Neural Representations (INRs) for decoding arbitrary resolutions. Given INRs' promise in TO [34, 35], we are inspired by Image Neural Field Diffusion (INFD) [8]. INFD introduces Convolutional Local Image Functions (CLIFs) for a high-quality, resolution-free autoencoder with finite latents, relaxing INR's pixel-independence. Our approach extends this idea beyond variable resolution to arbitrary aspect ratios and domain geometries, integrating a conditional latent diffusion model tailored for physics-based topology optimization.

**Representations of TO Problems.**   Many prior DL TO works used image-based boundary conditions and force representations [14, 20, 30, 33]. These often require FEA simulations, limiting FEA-free inference, or offer lower performance if FEA-free [14], and invariably suffer from resolution/shape dependence. Nobari et al. [34] proposed the Boundary Point Order-invariant MLP (BPOM), a point-cloud boundary condition representation that matches FEA-based input performance without requiring FEA simulation (See in Figure 2) and mitigating the resolution/shape dependence issues. We adopt BPOM for its generalizability and efficiency.

## 4   Methodology

Our method, OAT, a Latent Diffusion Model (LDM) [40], first employs an autoencoder to map topologies to fixed-dimension latents (decoded by a neural field for arbitrary resolution), then a conditional diffusion model generates topologies from these latents based on TO problem specifications (Figure 2; Appendix F for implementation details).

### 4.1   Resolution- and Shape-Agnostic Autoencoder with Neural Fields.

To handle the variability of mixed resolutions and aspect ratios (Fig. 2), we employ an autoencoder with a neural field renderer, inspired by INFD [8]. The encoder $E$ maps topology $T$ to latent $z = E(T)$; decoder $D$ converts $z$ to a feature tensor $\phi = D(z)$; and renderer $R$, extending the CLIF concept [8], reconstructs $\tilde{T} = R(\phi, c, s)$ using $\phi$, coordinates $c$, and cell/pixel sizes $s$.

**Scalable Training and Inference.**   Inspired by VQ-VAE [60] in LDMs [40], our autoencoder pads and resizes variable topologies to fixed $256 \times 256$ inputs for encoder $E$ (Fig. 2). Decoder $D$ then generates fixed resolution feature tensor $\phi$. A key challenge in TO is high-frequency changes with small changes in boundary conditions. To address this, we employ a convolutional renderer $R$ (unlike point-wise models [7, 34]) for better high-frequency details and scale consistency [8].

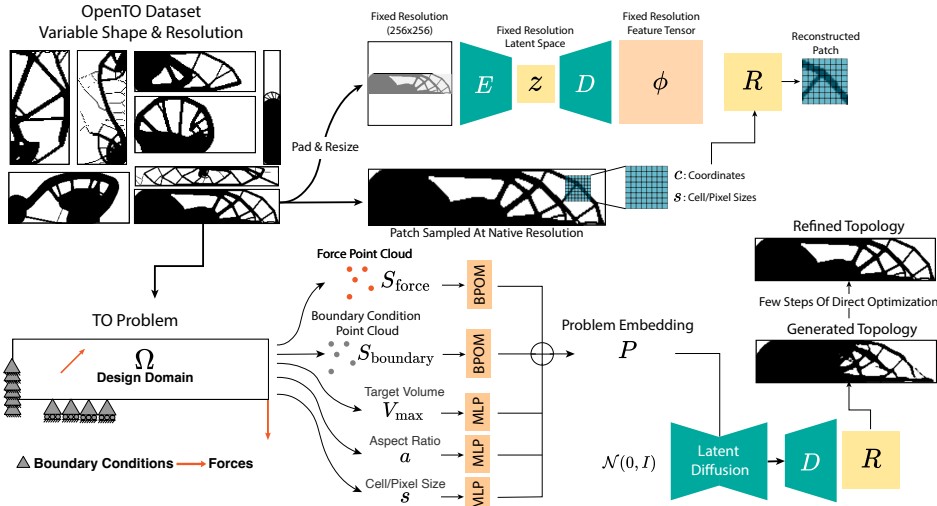

**Figure 2:** OAT generative framework overview. A resolution-free autoencoder encodes variable OpenTO dataset topologies into a fixed-resolution latent space. Latent diffusion models (LDMs) then conditionally generate topologies. Problem specifications, including forces and boundary conditions represented as point clouds, are processed by BPOM models and MLPs to form a fixed-size embedding to condition the LDM.

$R$'s local operator (inputs: nearest $\phi$ pixel, coordinates, cell sizes; Fig. 2) requires connected patch sampling given its convolutional nature. Thus, scalable training is possible with random patches from non-padding areas (Fig. 2). High-resolution inference is also possible by stenciled, overlapping padded tiles (padding $> R$'s receptive field) when memory is limited.

**Training Objective.** This neural field design yields a resolution and shape-agnostic autoencoder that gives us fixed-resolution latent representations. We train this autoencoder to minimize reconstruction error. For a ground truth patch or topology $T_{\text{GT}}$ (with pixel positions $c$ and cell sizes $s$) and its reconstruction $\tilde{T} = R(\phi, c, s)$, the objective is to minimize $L_1$ norm, which has proven more effective for reconstruction in prior works [7, 8]: $\mathcal{L}_{\text{AE}} = \|\tilde{T} - T_{\text{GT}}\|_1$.

## 4.2 Problem Representation

In topology optimization, a key challenge is to represent the underlying topology optimization problem, which includes domain shape, boundary conditions, and loads. This is difficult, as the number and position of these problem settings change in every configuration. We represent the TO problem (Eq. 8) by encoding its primary components using different resolution and shape-independent encoders. The problem domain, $\Omega$ (rectangular grid with regular cells in our dataset), is described by its aspect ratio $a \in \mathbb{R}^2$ and its resolution via cell/pixel size $s$ (Fig. 2). The target material volume, $V_{\text{max}}$, is represented as a scalar volume fraction $VF = V_{\text{max}}/V_{\Omega}$, which indicates the ratio of allowed material to total volume of the domain $V_{\Omega}$. The more complex components, forces, and boundary conditions are handled as point clouds. Boundary conditions, which fix specific degrees of freedom, are represented as a point set $S_{\text{boundary}}$ with binary features indicating directional fixture (Fig. 2). Similarly, applied forces are a point set $S_{\text{force}}$ with 2D vector features denoting load magnitude and direction (Fig. 2). As these sets are order invariant, we employ the Boundary Point Order-invariant MLP (BPOM) approach [34] to learn the embeddings of $S_{\text{boundary}}$ and $S_{\text{force}}$. The complete latent problem embedding, $P$, is a concatenation of five embeddings from distinct neural networks:

$$P = \text{concat}(\text{BPOM}_{\text{b}}(S_{\text{boundary}}), \text{BPOM}_{\text{f}}(S_{\text{force}}), \text{MLP}_{\text{vf}}(VF), \text{MLP}_{\text{cell}}(s), \text{MLP}_{\text{ratio}}(a)), \quad (4)$$

where $\text{BPOM}_{\text{b}}$ and $\text{BPOM}_{\text{f}}$ embed boundary conditions and forces, respectively, while Multi-Layer Perceptrons (MLPs) embed the volume fraction ($VF$), cell size ($s$), and aspect ratio ($a$). This construction yields a fixed-size embedding $P$ capable of representing TO problems with diverse domain shapes, resolutions, and arbitrary boundary conditions and forces. For brevity, we denote this entire encoding process as $P = \text{LP}(\hat{P})$, where $\hat{P}$ encompasses all raw problem inputs.

## 4.3 Latent Diffusion and Classifier-Free Guidance

Having trained the autoencoder, we obtain latent representations $z = E(T)$ for each topology $T$. Furthermore, the TO problem is encoded into a latent problem embedding $P = \text{LP}(\hat{P})$. We then train our diffusion model to predict the 'velocity' conditioned on the problem definition, $\hat{P}$, and train it with the following modified diffusion objective:

$$\mathcal{L}_{\text{LDM}} = \mathbb{E}_{z \sim E(T), t, \epsilon \sim \mathcal{N}(0, I)} \left[ \left\| v - v_\theta \left( z_t, t \mid \hat{P} \right) \right\|^2 \right]. \tag{5}$$

where $v = \sqrt{\bar{\alpha}_t} \epsilon - \sqrt{1 - \bar{\alpha}_t} z$. To enable classifier-free guidance [17], the problem inputs $\hat{P}$ are omitted for 50% of training samples, allowing the model to learn both conditional and unconditional denoising. This allows us to control the influence of conditioning during inference as described by Ho and Salimans [17]. Classifier-free guidance can be applied in our approach by adjusting the denoising velocity. Normally, in conditional denoising, one would use the predicted velocity at time step $t$, $v_\theta(z_t, t \mid P)$, which predicts the velocity for the current noisy sample $z_t$ given problem embedding $P$. When applying classifier-free guidance we can also compute the un-conditional denoising, namely $v_\theta(z_t, t \mid \varnothing)$, and rather than just using $v_\theta(z_t, t \mid P)$, we measure the shift in velocity direction as a result of conditioning, $v_\theta(z_t, t \mid P) - v_\theta(z_t, t \mid \varnothing)$ and amplify this shift by an over-relaxation factor, $\omega$, leading to a final denoising velocity $\hat{v}_\theta(z_t, t, P) = v_\theta(z_t, t \mid \varnothing) + w \left( v_\theta(z_t, t \mid P) - v_\theta(z_t, t, \varnothing) \right)$, which can be used in place of $v_\theta(z_t, t \mid P)$, mimicking what Ho and Salimans [17] propose for classifier-free guidance.

## 4.4 Post Generation Optimizer Refinement.

Despite the ever-increasing quality of DGM-generated solutions and other deep learning methods for TO, we observe that these models often lack precision in many cases and can lead to failed designs. To address this, we follow the few-step direct optimization approach commonly used in literature [14, 34]. A few optimization steps (5-10) of SIMP (small compared to 200-500 iterations for full optimization) are equivalent to a local search operation and are applied to topologies synthesized by OAT to improve the accuracy of the final samples. We also incorporate this into our framework; however, as we discuss in our experiments, even without any direct optimization, OAT outperforms other models that require optimization steps.

## 4.5 OpenTO Dataset

Existing datasets in DL for TO face four challenges that we have overcome in creating the OpenTO dataset: 1) Most existing datasets are limited to one specific domain shape (square) and one resolution (64 x 64), with the most expansive dataset including five different shapes and resolutions [34]; 2) Most existing works are limited to a small set of pre-defined boundary conditions (a maximum of 42 in 2D dataset [34]); 3) Existing datasets typically feature only a single applied force; 4) All existing datasets only apply forces and boundary conditions at the borders of the design space (boundary of domain), and lack interior forcing and fixtures.

These limitations restrict the development of foundation models and are a key factor behind the generalizability challenges. Thus, we see this as a major gap and introduce the largest open-source Topology Optimization (**OpenTO**) dataset for *general purpose* topology optimization with 2.194M samples. To enable a foundational scale of data, OpenTO comprises procedurally-created random TO problems, running a SIMP solver, and obtaining optimal topologies from the solver. The data generation is done such that OpenTO overcomes the aforementioned limitations of prior works. The main features of what makes OpenTO effective are: 1) Design domains in OpenTO are fully random in both resolution and shape with aspect ratios ranging from very narrow (10 to 1), to square (1 to 1), and pixel/cell size ranging from 1/64 to 1/1024, covering the exhaustive range of rectangular domain shapes and resolution in most practical real-world applications of TO in 2D; 2) OpenTO includes randomly sampled boundary conditions, including **interior boundary condition point**, with every sample having a unique boundary condition configuration; 3) OpenTO includes fully random forces including interior forces, and configurations with as many as 4000 loads (see Appendix D on distributed forces) in one configuration. The exhaustive nature of OpenTO makes it the first dataset to tackle the general TO problem and a starting point for the development of foundation models for TO. OpenTO includes 5,000 test samples, with a fully randomized configuration not in the main training data for testing the performance of models in a general problem setting. The full details of the procedural data generation can be found in Appendix D.

Finally, OpenTO additionally includes the prior datasets developed in limited settings (194,000 samples from Nobari et al. [34]). Overall, OpenTO includes 2.194M samples of topologies, with 894,000 samples being labeled (with defined $\hat{P}$) and the rest only including the topology. We use all samples to train our autoencoder, and only use the 894,000 samples with labels to train our LDM.

## 5 Experiments

In this section, we perform experiments to showcase OAT's increased performance compared to prior works trained on the smaller, limited datasets. This constitutes the first truly general TO benchmark on fully randomized problems. Before detailing the experiments, we will first briefly describe the evaluation metrics we use to measure the performance of each model.

**Evaluation Metrics:**    As we described in Section 2.1, TO involves both a compliance objective and a volume fraction constraint. Thus, we measure the performance of models with respect to these two requirements of the TO problem. First, to measure how well the models perform in compliance minimization, we measure compliance error (CE), which is calculated by subtracting the compliance of a generated sample from the compliance of the SIMP-optimized solution for the corresponding problem, then normalizing by the SIMP-optimized compliance value. Since the mean compliance error tends to be dominated by just a few outlier samples, we also report the median compliance error. We also report mean volume fraction error (VFE), which quantifies the absolute error between the generated topology's actual volume fraction and the target volume fraction for the given problem. This benchmarking is standard practice for DGMs in TO [14, 30, 33, 34].

**Experimental Setup.**    Our experiments benchmark OAT both on existing benchmarks with limited boundary conditions and on our new OpenTO test set with generalized problem configurations. In all experiments, we train OAT exclusively on the OpenTO dataset, and never on any benchmark-specific data. On existing benchmarks, we compare against the *specialized* existing models trained specifically for these benchmarks and report performance metrics as reported by the original authors. On the OpenTO benchmark, only one prior work, NITO [34], has the architectural generalizability to be trained on this data. Thus, we train OAT with a combined 729.94M (672.00M for LDM and 57.94M for AE) parameters, training the diffusion model for 50 epochs (349,219 steps). For a fair baseline, we train a variant of NITO for the same number of training steps, scaled to 732M parameters to approximately match OAT. For each method, we then sample solutions for each problem and test performance with and without direct optimization. In all experiments, OAT sampling is done with 20 DDIM deterministic denoising steps with a classifier-free guidance scale of 2.0 (as informed by our ablation studies presented in Appendix B. We sample 10 sets of solutions for each problem and report the average across runs.

**Table 1:** Quantitative evaluation on 64 x 64 datasets. All models were only trained on 64 x 64 data, except OAT, which is trained on the OpenTO dataset. w/ G: using a classifier and regression guidance. CE: Compliance Error. VFE: Volume Fraction Error. + 5 and + 10 indicate 5 and 10 steps of post-generation direct optimization. OAT, without any post-generation optimization, outperforms NITO + 10, the SOTA model that also uses 10 steps of optimization.

| Model | CE % | CE % Med | VFE % |
|---|---|---|---|
| TopologyGAN [33] | 48.51 | 2.06 | 11.87 |
| cDDPM [14] | 60.79 | 3.15 | 1.72 |
| TopoDiff [34] | 3.23 | 0.45 | 1.14 |
| TopoDiff w/ G [34] | 2.59 | 0.49 | 1.18 |
| DOM w/o TA [14] | 13.61 | 1.79 | 1.86 |
| DOM w/ TA [14] | 4.44 | 0.74 | 0.74 |
| NITO [34] | 8.13 | 0.47 | 1.40 |
| **OAT (Ours)** | **1.74** | **0.32** | **0.25** |
| NITO + 5 [34] | 0.30 | 0.12 | 0.40 |
| TopoDiff + 5 [34] | 3.55 | 0.42 | 0.67 |
| TopoDiff w/ G + 5 [34] | 2.24 | 0.44 | 0.69 |
| **OAT (Ours)** + 5 | **0.22** | **0.041** | **0.39** |
| NITO + 10 [34] | 0.17 | 0.071 | **0.25** |
| TopoDiff + 10 [34] | 1.38 | 0.33 | 0.45 |
| TopoDiff w/ G + 10 [34] | 1.05 | 0.32 | 0.45 |
| **OAT (Ours)** + 10 | **0.0503** | **0.014** | 0.27 |

### 5.1 Comparison To State of The Art

Prior works are typically limited to specific domain shapes and resolutions. We first compare the performance of OAT with prior state-of-the-art on their own test sets [30, 33, 34, 34].

**The 64 x 64 Benchmark.**    The most common benchmark that previous state-of-the-art models [14, 33, 33, 34] have trained on and benchmarked against is TO problems with a single force, and only 42 distinct boundary condition configurations defined on a 64 x 64 domain. We evaluate OAT on this benchmark with no specific training or fine-tuning. We also test post-generation optimization, which some prior works [14, 34] explore. Table 1 shows the results on the 64 x 64 benchmark. We can see that OAT has the lowest compliance error amongst models without direct optimization, with only 1.74% error compared to 2.59% from the best model (TopoDiff with classifier guidance), showing a 32% reduction in compliance error. Notably, OAT achieves this while maintaining stricter adherence to the target volume fraction, demonstrating superior efficiency in material usage. Table 1 shows that when post-generation optimization is applied, OAT continues to outperform all competing methods,

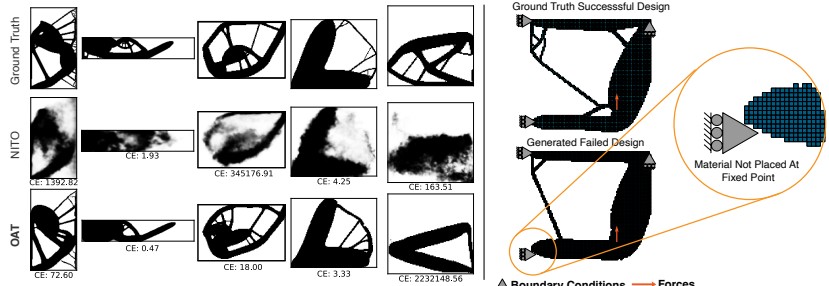

**Figure 3:** Left: Samples from ground truth (top), NITO (middle), and OAT (bottom) on a random OpenTO benchmark; compliance errors shown below. NITO outputs are noisy with many gray pixels, unsuitable for real-world use. Right: Example highlighting sensitivity in TO; slight deviations by OAT led to material misplacement at boundary conditions, causing design failure. This highlights the precision requirements of TO.

achieving the lowest compliance and volume-fraction errors across all settings. This represents the first instance in topology optimization where a foundation-model-based approach not only generalizes across problem configurations but also surpasses all specialized, task-specific models trained on restricted datasets. This establishes OAT as a major milestone in advancing generalizable, data-driven topology optimization.

**The 256 x 256 Benchmark.** Nobari et al. [34], in their neural field-based method, expand the prior mentioned data by solving the same problems at a much higher resolution of 256 x 256 resolution (still the same boundary conditions and single load) and test their method and the highest performing prior work, TopoDiff [30], on this data by retraining a larger TopoDiff on this resolution. We report the results of our model and the reported values for NITO and TopoDiff [34], in Table 2. The trends here are similar to the 64 x 64 data, with an even larger gap in performance, with specialized models unable to beat the general framework of OAT in this harder high resolution benchmark. However, we do see that NITO, which predicts blurry gray samples [34], allows more freedom for direct optimization, thus yielding marginally better compliance errors with few-step optimization. However, note that OAT can perform nearly on par with these costly direct optimization methods without using any direct optimization.

**Extended five shape benchmark.** Aside from extending the dataset to higher resolution, Nobari et al. [34] also expands the data by introducing three additional domain shapes apart from 64 x 64 and 256 x 256. The authors add data for 64x48, 64x32, and 64x16 domains, still with the same boundary condition configuration adapted to these new shapes. The authors then train their model on all of this data and report the performance on this benchmark. Unlike prior works, which could not handle this kind of mixed shape/resolution data, our approach, like NITO [34], can be adapted to this data easily. Table 3 shows how a NITO model only trained on this dataset lags behind OAT, as we saw in prior benchmarks.

**Table 2:** Quantitative evaluation on 256 x 256 datasets. Same settings as 64 x 64.

| Model | CE % | CE % Med | VFE % |
|---|---|---|---|
| TopoDiff [34] | 16.62 | 0.59 | 2.92 |
| NITO [34] | 9.178 | 0.96 | 1.52 |
| **OAT (Ours)** | **1.51** | **0.51** | **-0.17** |
| NITO + 5 [34] | **0.25** | **0.09** | 0.34 |
| **OAT (Ours)** + 5 | 0.54 | 0.37 | **-0.12** |
| NITO + 10 [34] | **0.033** | **0.012** | 0.130 |
| **OAT (Ours)** + 10 | 0.081 | 0.16 | **-0.0502** |

**Table 3:** Quantitative evaluation on the combined 5 shape datasets. NITO results are only available with direct optimization in the original work [34].

| Model | CE % | CE % Med | VFE % |
|---|---|---|---|
| NITO + 10 [34] | 0.56 | 0.13 | 0.39 |
| **OAT (Ours)** | 1.94 | 0.89 | 0.103 |
| **OAT (Ours)** + 5 | 0.82 | 0.12 | 0.35 |
| **OAT (Ours)** + 10 | **0.23** | **0.091** | **0.28** |

## 5.2 The General Benchmark

The only prior work that can handle the OpenTO dataset is the NITO [34] model, which is a non-generative neural field model. As such, we train NITO on OpenTO and report the performance of the model on our fully random benchmark. Notably, even though we train a variant of NITO with the same number of parameters, we observe very poor visual quality in samples generated by NITO (Figure 3). The authors of the original work had also observed non-sharp boundaries and suggested that this issue [34] may be due to the deterministic (non-generative) nature of their model. OAT learns this more complex benchmark significantly better. Figure 3 shows that the quality of samples generated by OAT is more realistic. Nonetheless, OpenTO presents a challenging benchmark that both OAT and NITO struggle with.

**Table 4:** Quantitative evaluation on the fully general benchmark of TO problems. It can be seen that OAT outperforms NITO-L, which is the only existing model that could be trained on OpenTO. * Values are reported for the non-failure samples.

| Model | CE* % | CE* % Med | VFE* % | Failure Rate % |
|---|---|---|---|---|
| NITO | 13.14 | 6.43 | 5.3861 | 60.01 |
| **OAT (Ours)** | **8.17** | **3.75** | **-0.12** | **39.22** |
| NITO + 5 | 5.10 | 1.78 | 1.75 | 23.88 |
| **OAT (Ours) + 5** | 6.29 | 2.86 | 0.68 | 22.11 |
| NITO + 10 | 3.56 | 1.33 | 0.83 | 16.39 |
| **OAT (Ours) + 10** | 4.31 | 2.24 | 0.57 | 15.90 |

**Table 5:** Inference time scaling results of generating multiple samples and reporting the results for the best of $N$ samples generated by OAT on the fully random benchmark.

| Best of | CE* % | CE* % Med | VFE* % | Failure Rate % |
|---|---|---|---|---|
| 2 | 7.40 | 3.02 | -0.0021 | 32.95 |
| 4 | 5.50 | 2.65 | 0.041 | 27.50 |
| 8 | 4.14 | 2.41 | 0.096 | 21.50 |
| 16 | 1.97 | 1.58 | 0.14 | 16.15 |
| 32 | 1.76 | 1.55 | 0.15 | 12.92 |
| 64 | 0.4439 | 1.31 | 0.19 | 11.51 |

**Failure Study.** Given the precise nature of the topology optimization problem, small changes in material distribution can cause catastrophic failures in the outcome. This often occurs when the model fails to place material on boundary conditions or loads precisely. The example in Figure 3 shows how a failed topology looks very similar to the ground truth topology but has poor physical performance. Since DGMs often focus on distribution matching but fail to capture the precise requirements of the problem, such failures are common in DL-based TO. In this benchmark, we define failure as having a compliance error greater than 100%, meaning that the generated topology has a compliance more than double that of the conventional optimizer. Despite OAT frequently incurring such failures, its failure rate is still significantly lower than NITO's, as shown in Table 4. Given that the topologies in most failure cases only need small corrections, 5 to 10 steps of optimization reduce OAT's failure rate from 39% to only 16%. This confirms that these imprecisions can be healed with only a handful of optimization steps. Further improvements to failure rate may be addressed by larger models and datasets, as well as more advanced paradigms emerging in foundation model training, such as reinforcement learning techniques [36, 47] or utilizing invalid data [39].

**Generalized Performance.** In addition to the failure rate, we also report the compliance and volume fraction errors for each method with and without direct optimization, measured only on the successful samples. OAT significantly outperforms NITO in both of these physics-based performance metrics. Notably, OAT has a compliance error of less than 10% on the fully random general data, which reduces to only 5% after a few steps of optimization. Comparing this to the first models for deep learning in TO, such as TopologyGAN [33], which had over 50% error on the limited 64 x 64 dataset, a 5% error on fully random data marks a significant step forward for the community and heralds a potential paradigm shift toward foundation models.

**Table 6:** Average inference time at 64 x 64 and 256 x 256 resolutions on an RTX 4090 GPU (10-run average). Some timings include 10 refinement iterations (marked w/ Ref). NITO-S and NITO-L are original and larger variants [34]; OAT timings use 20 DDIM steps. Parentheses show an increase factor when elements scale by 16x. SIMP times are reported using an Intel 14-900K CPU (RTX4090 for SIMP-GPU). All SIMP inference speeds are measured for 150 steps of optimization (average needed to converge).

| Model | Parameters (M) | Inference Time (s) |
|---|---|---|
| *64 x 64 Resolution* | | |
| TopoDiff | 121 | 1.86 |
| TopoDiff w/ G | 239 | 4.79 |
| DOM | 121 | 0.82 |
| SIMP | - | 3.45 |
| SIMP GPU | - | 9.13 |
| NITO-S | 22 | 0.005 |
| NITO-S w/ Ref | 22 | 0.14 |
| NITO-L | 732 | 0.05 |
| NITO-L w/ Ref | 732 | 0.18 |
| **OAT (Ours)** | 730 | 0.508 |
| **OAT (Ours) w/ Ref** | 730 | 0.637 |
| *256 x 256 Resolution* | | |
| TopoDiff | 553 | 10.81 (5.812×) |
| TopoDiff w/ G | 1092 | 22.04 (4.601×) |
| DOM | 553 | 7.82 (9.537×) |
| SIMP | - | 69.45 (20.13×) |
| SIMP GPU | - | 68.30 (19.80×) |
| NITO-S | 22 | 0.16 (32.00×) |
| NITO-S w/ Ref | 22 | 2.88 (20.57×) |
| NITO-L | 732 | 0.67 (13.40×) |
| NITO-L w/ Ref | 732 | 3.52 (19.56×) |
| **OAT (Ours)** | 730 | 0.51 (**1.003×**) |
| **OAT (Ours) w/ Ref** | 730 | 3.28 (**5.129×**) |

**Inference Time Scaling Study.** The high failure rate observed in Table 4 may seem stark upon first glance; however, we believe that the generative nature of OAT provides a clear path to alleviate this issue. With generative models like OAT, one can generate vast numbers of candidates in a short amount of time, considering batch inference of diffusion models scales better than sampling one

problem at a time. In Table 5, we generate multiple candidates for each problem and measure the performance of the best amongst all samples. These results clearly show how the failure rate goes down rapidly as more samples are evaluated, with as few as eight samples yielding only 20% failure and a CE that is half that of the single sample run in Table 4. We believe this shows how powerful foundational generative frameworks like OAT can be for physical AI. Moreover, this favorable scaling with more samples clearly shows how emerging approaches around reinforcement learning [36, 46] in generative models can be employed here to further improve the performance of OAT.

### 5.3  Sampling Efficiency & Inference Speed

Inference speed is one of the key advantages of deep learning over iterative optimization. It's important to mention that a high-level analysis of computational cost in this case would result in an $O(N)$ complexity for OAT, where $N$ is the number of pixels/elements for a given sample. This is because the non-constant cost of OAT is in the neural field renderer, which would have a complexity of $O(N)$. This is compared to the expensive FEA simulations at each step of the optimization, yielding an $O(N^3)$ complexity (Solving a linear system of equations). However, given that FEA matrices in this setting are highly sparse and the meshes in our work are structured, specialized multi-grid solvers can, in theory, reach an $O(N)$ efficiency [1, 28, 58]. This high-level analysis is however, fails to realize the inference efficiency of deep learning models compared to conventional optimization, given that it does not take into account the highly parallelized inference of deep learning models on the GPU compared to the iterative nature of linear system solvers. This makes inference time an overall better measure of performance in practical consumer hardware. Thus, it is important to analyze the inference time of different approaches. Table 6 shows the results of inference speed for different models. NITO, the only model with comparable generalizability, is faster at 64 x 64 resolution, but requires the whole network for each sample point over the entire field, meaning that it scales poorly with resolution. In contrast, the latent diffusion model, whose autoencoder only has 40M parameters, can be sampled at high resolution much more efficiently. NITO heavily relies on post-sampling optimization, which often contributes the majority of inference cost. Even ignoring this optimization case, OAT scales significantly faster than NITO and generates samples much faster at higher resolutions, despite having the same number of parameters. OAT even outperforms similar diffusion models in inference speed. For example, compared to TopoDiff's 100 denoising steps, OAT's DDIM sampling requires only 20 denoising steps, significantly accelerating inference. Most importantly, unlike some prior diffusion models, OAT remains faster than optimization even at low resolution, where direct optimization is relatively fast. Overall, OAT is significantly faster than SIMP, making it an attractive option for fast design space exploration.

It is important to note that some critics of generative models for TO point to the break-even analysis of computational cost [62]. We estimate this and discuss this concern in more detail in Appendix G.

## 6  Conclusion & Future Work

We have introduced *Optimize Any Topology* (OAT), the first foundation-model approach to structural topology optimization that is agnostic to domain shape, aspect ratio, and resolution. Trained on the new OPENTO corpus comprising 2.2 million optimized designs, OAT couples a resolution-free latent auto-encoder with a conditional diffusion prior and achieves much lower mean compliance error than prior deep-learning baselines on the canonical $64 \times 64$ benchmark (1.7 % versus 2.59 %) while delivering sub-second inference. Moreover, OAT is easily extendable to different TO problems and physics, such as heatsink optimization, given OAT's auto-encoder has great zero-shot capability for reconstructing such topologies without re-training (see Appendix C). This clearly solidifies the case for OAT as a foundational framework for TO. On a fully general benchmark with random boundary conditions and resolutions, OAT attains $< 10\,\%$ mean compliance error. These results demonstrate that large-scale pre-training combined with resolution-free latent diffusion is a viable path towards real-time, physics-aware design exploration. Despite the promising performance of OAT, significant challenges remain. First, we note that OAT faces a notable failure rate on fully random benchmarks, which future work will focus on addressing through directions such as reinforcement learning [36, 46] based on optimizer guidance in diffusion models [11]. Future work should also focus on addressing multi-physics objectives, and develop few-shot fine-tuning approaches to quickly adapt OAT or other foundational frameworks to new physics such as stress-constrained and buckling TO problems.

## 7  Acknowledgements

We would like to acknowledge Akash Srivastava and the MIT-IBM Watson AI Labs for helpful insights and guidance in the early stages of developing our methods. We would also like to acknowledge the crucial guidance from Professor Josephine Carstensen in developing the solver used to generate the data and perform benchmarks in this work. PI Ahmed acknowledges support from the National Science Foundation under CAREER Award No. 2443429.

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

# Table of Contents for Appendices

# A  Topology Optimization and The Minimum Compliance Problem

Structural Topology Optimization (TO) is the process of determining the optimal distribution of a limited amount of material within a defined design space to maximize (or minimize) specific physics-based performance metrics. Let $\Omega \subset \mathbb{R}^d$ represent a bounded design domain with boundary $\Gamma = \partial\Omega$. The system's physical behavior is described by a state field $u : \Omega \to \mathbb{R}^m$ (e.g., displacement), which is the solution to a governing partial differential equation (PDE). For simplicity, considering only Dirichlet boundary conditions, this PDE takes the form:

$$\mathcal{L}\big(\rho(x),\, u(x)\big) = f(x) \quad \text{in } \Omega, \qquad u(x) = g(x) \quad \text{on } \mathcal{D} \subset \overline{\Omega} \tag{6}$$

Here, $\rho(x) : \Omega \to [0, 1]$ is the material density at each point $x$, acting as the design variable we seek to optimize. $\mathcal{L}$ is a differential operator representing the underlying physics (e.g., elasticity), $f(x)$ represents applied forces or sources, and $g(x)$ prescribes values for $u(x)$ on a portion of the domain $\mathcal{D}$ (which can be on the boundary $\Gamma$ or within $\Omega$). The general continuous TO problem is to find the material density distribution $\rho(x)$ that minimizes a performance objective $J\big(u(\rho),\, \rho\big)$, such as structural compliance (inverse of stiffness) or thermal resistance. This is subject to the governing PDE (Eq. 6) and a constraint on the total material volume $V_{\max}$:

$$\min_{\rho(\cdot)}\ J\big(u(\rho),\, \rho\big) \quad \text{s.t.} \quad \mathcal{L}\big(\rho, u\big) = f(x),\ u(x) = g(x) \quad \text{on } \mathcal{D}\ , \int_\Omega \rho \,\mathrm{d}x \le V_{\max}. \tag{7}$$

This process and the overall problem are depicted in Figure 1. While the ideal outcome is a distinct structure $W \subset \Omega$ with clear boundaries (binary densities), optimizing for such a structure directly is challenging. Furthermore, analytical (closed-form) solutions to the PDE in Eq. 6 are generally unobtainable for complex geometries and material distributions. These practical difficulties necessitate a numerical approach to TO. This is usually done by discretizing the domain into a mesh with finite elements/cells, and solving the problem in this discretized space using Finite Element Analysis (FEA) [9].

So far, we described the general topology optimization problem in continuous space; however, as we alluded to, this problem is often not approached in the continuous form. To make the problem tractable, the continuous domain $\Omega$ is discretized into a finite element mesh $\mathcal{T}_h$, composed of $N$ small elements $\mathcal{T}_h = \{K_e\}_{e=1}^N$. Within this framework, the continuous material density $\rho(x)$ is approximated by a vector of element densities $\rho_h = [\rho_1, \ldots, \rho_N]^\top$, where each $\rho_e \in [0, 1]$. Similarly, the state field $u(x)$ becomes a vector $u_h = [u_1, \ldots, u_M]^\top$ representing values at $M$ discrete points or "degrees of freedom" (DoFs) of the mesh (e.g., 2D displacements at nodes). Applied forces $f(x)$ and prescribed boundary values $g(x)$ are also discretized into vectors $f_h$ and $g_h$, where the boundary conditions are applied at specific DoFs $\mathcal{D}_h$. The local element operator (e.g., stiffness matrix) $K_e$ is derived from $\mathcal{L}$ for each element [9]. The numerical TO problem then becomes finding the optimal discrete densities $\rho_h$:

$$\min_{\rho_h \in [0,1]^N}\ J_h\big(u_h(\rho_h), \rho_h\big) \quad \text{s.t.} \quad \begin{cases} K(\rho_h)\, u_h = f_h, \\ \displaystyle\sum_{e=1}^N v_e\, \rho_e \le V_{\max}, \\ u_i = g_i, \quad i \in \mathcal{D}_h, \end{cases} \tag{8}$$

where $J_h$ is the discretized objective function, $K(\rho_h)$ is the global system matrix (e.g., global stiffness matrix) assembled from element contributions, which depends on the material densities $\rho_h$, and $v_e$ is the volume of element $e$. The first constraint, $K(\rho_h)u_h = f_h$, is the discretized form of the governing PDE (Eq. 6). Although the goal is often a binary design ($\rho_e \in \{0, 1\}$, indicating presence or absence of material), directly solving this as a mixed-integer non-linear program is computationally prohibitive for realistic problem sizes. Therefore, the optimization typically allows continuous densities $\rho_e \in [0, 1]$ and employs penalization techniques to encourage solutions that are nearly binary. A widely used method is Solid Isotropic Material with Penalization (SIMP) [3, 41]. In SIMP, effective element densities are related to design variables $\phi_e \in [0, 1]$ by $\rho_e = \phi_e^p$, where $p > 1$ is a penalization factor that makes intermediate density values (between 0 and 1) structurally inefficient, thus favoring $\phi_e$ values close to 0 or 1.

In our work, we focus on the minimum compliance problem, which involves solving the linear elasticity problem and using an FEA solver and iteratively optimizing the topology in discrete space. We detail this in the section below.

### A.1 Minimum Compliance Topology Optimization

This section details a prominent application of Topology Optimization (TO): minimum compliance structural optimization, often referred to as stiffness optimization. The primary objective is to determine the material layout that results in the stiffest possible structure under applied mechanical loads, subject to a constraint on the total amount of material used. We will first describe the underlying physics model for this class of problems.

### A.1.1 The Physics Model: Static Linear Elasticity

In most structural topology optimization problems, including minimum compliance, the behavior of a structure under load is modeled using linear elasticity. This model assumes that the structure is composed of a material that exhibits linear elastic behavior (i.e., stress is proportional to strain via Hooke's Law) and that deformations are small.

The governing partial differential equation (PDE) for linear elasticity in its general dynamic form is the Navier-Cauchy equation:

$$\frac{E}{2(1+\nu)}\nabla^2\mathbf{u} + \left(\frac{E\nu}{(1+\nu)(1-2\nu)} + \frac{E}{2(1+\nu)}\right)\nabla(\nabla \cdot \mathbf{u}) + \mathbf{f}_{body} = \rho_{phys}\ddot{\mathbf{u}}, \tag{9}$$

where $\mathbf{u}(x,t)$ is the displacement vector field (corresponding to $u(x)$ in Eq. 6, where $m$ would be the number of spatial dimensions, e.g., $m = 2$ or $m = 3$), $\ddot{\mathbf{u}}$ is its second derivative with respect to time (acceleration), $\mathbf{f}_{body}(x)$ is the body force vector field (e.g., gravitational loads, corresponding to $f(x)$ in Eq. 6 if $\mathcal{L}$ is defined as the internal force operator), $E$ is the Young's modulus (a measure of material stiffness), $\nu$ is the Poisson's ratio (characterizing transverse contraction/expansion), and $\rho_{phys}$ is the physical mass density of the material.

For static or quasi-static analyses, which are typical in minimum compliance problems, we are interested in the steady-state deformation under load. This allows us to set the inertial term $\rho_{phys}\ddot{\mathbf{u}}$ to zero. The governing PDE for static linear elasticity then becomes:

$$\frac{E}{2(1+\nu)}\nabla^2\mathbf{u} + \left(\frac{E\nu}{(1+\nu)(1-2\nu)} + \frac{E}{2(1+\nu)}\right)\nabla(\nabla \cdot \mathbf{u}) + \mathbf{f}_{body} = \mathbf{0}. \tag{10}$$

This equation, along with appropriate boundary conditions (such as prescribed displacements $g(x)$ on $\mathcal{D}$ as in Eq. 6), defines the structural response. In the context of TO, the Young's modulus $E$ is not uniform throughout the domain $\Omega$; instead, it varies spatially depending on the material density distribution $\rho(x)$, which is the design variable.

### A.1.2 Finite Element Discretization and Compliance Calculation

As discussed in the general TO framework (leading to Eq. 8), analytical solutions to Eq. 10 are generally intractable for complex domains and material distributions. Therefore, the Finite Element Method (FEM) is employed. The continuous domain $\Omega$ is discretized into a mesh $\mathcal{T}_h$ of $N$ elements, and the continuous displacement field $\mathbf{u}(x)$ is approximated by a vector of $M$ discrete nodal displacements $u_h \in \mathbb{R}^M$. The body forces $\mathbf{f}_{body}(x)$ and any applied surface tractions are discretized into a global force vector $f_h \in \mathbb{R}^M$. Dirichlet boundary conditions (supports) are enforced by prescribing values for certain components of $u_h$.

The FEM discretization of Eq. 10 results in a system of linear algebraic equations:

$$K(\rho_h)u_h = f_h, \tag{11}$$

where $K(\rho_h) \in \mathbb{R}^{M \times M}$ is the global stiffness matrix. This matrix depends on the vector of element design variables (densities) $\rho_h = [\rho_1, \ldots, \rho_N]^\top$, as these densities determine the material properties (specifically, Young's modulus) within each element.

The compliance $C$ of the structure is a measure of its overall flexibility; minimizing compliance is equivalent to maximizing stiffness. For the discrete system, compliance is calculated as the work done by the external forces:

$$C(u_h, f_h) = f_h^T u_h. \tag{12}$$

Using the equilibrium condition $f_h = K(\rho_h)u_h$, compliance can also be written as $C = u_h^T K(\rho_h)u_h$.

### A.1.3 Material Interpolation: The SIMP Model

To enable optimization, the material properties of each element must be related to the design variables $\rho_e$. The new paper's $\rho_e \in [0, 1]$ are the design variables for each element $e$. A common approach is the Solid Isotropic Material with Penalization (SIMP) method. In this scheme, the Young's modulus $E_e$ of an element $e$ is interpolated based on its normalized density $\rho_e$:

$$E_e(\rho_e) = E_{min} + \rho_e^p(E_{solid} - E_{min}), \tag{13}$$

where $E_{solid}$ is the Young's modulus of the solid material, $E_{min}$ is a small positive Young's modulus assigned to void regions (e.g., $\rho_e \approx 0$) to prevent the stiffness matrix $K(\rho_h)$ from becoming singular, and $p$ is the penalization factor, typically $p \geq 3$. This penalization makes intermediate densities (e.g., $\rho_e = 0.5$) structurally inefficient, meaning they contribute less to stiffness than their "cost" in material. This encourages the optimization process to yield designs with densities close to 0 (void) or 1 (solid). The element stiffness matrices $K_e$ are computed using $E_e(\rho_e)$ and then assembled into the global stiffness matrix $K(\rho_h)$. Note that the choice of $E_{min}$ is purely to avoid singular values and is selected to be $10^-9$ in our data generation compared to the much larger value of $E_{\text{solid}} = 1.0$. This choice is informed by prior literature recommending such a value as a good balance between stable optimization and accuracy of simulations [50]. Since in our implementation we set $E_{\min} = 10^{-9}$, the choice of $\rho_{\min} = 0$ is used, although numerically any design element in a mesh which receives a zero density will effectively have a Young's modulus of $10^{-9}$ which given our choice $E_{\text{solid}} = 1$ would be equivalent to setting $E_{\min} = 0$ and $\rho_{\min} = 10^{-9}$.

### A.1.4 The Minimum Compliance Optimization Problem

The goal of minimum compliance topology optimization is to find the distribution of material densities $\rho_h$ that minimizes the compliance $C$ (Eq. 12), subject to the static equilibrium constraint (Eq. 11) and a constraint on the total volume of material used. This is a specific instance of the general discretized TO problem (Eq. 8). The formulation is:

$$
\begin{aligned}
\underset{\rho_h}{\text{minimize}} \quad & C(u_h, f_h) = f_h^T u_h \\
\text{subject to:} \quad & K(\rho_h)u_h = f_h && \text{(Static Equilibrium)} \\
& \sum_{e=1}^{N} v_e \rho_e \leq V_{\max} && \text{(Volume Constraint)} \\
& \rho_{min} \leq \rho_e \leq 1 && \text{for } e = 1, \dots, N \text{ (Density Bounds)},
\end{aligned} \tag{14}
$$

where $v_e$ is the volume of element $e$, $V_{\max}$ is the maximum permissible total volume of the material (as defined in Eq. 7 and Eq. 8), and $\rho_{min}$ is a small positive lower bound for the element densities (e.g., $10^{-9}$ or $10^{-6}$) to ensure $E_e$ is always positive if $E_{min} = 0$ is chosen, or to represent a minimum manufacturable material thickness. This formulation is widely used in structural optimization. The aim of approaches like Neural Network based Inverse Topology Optimization (NITO), as alluded to in your previous work, would be to predict the optimal densities $\rho_h$ directly, given the problem definition (loads $f_h$, boundary conditions represented in $K$ and $f_h$, and the target volume fraction $V_{\max}/\sum v_e$).

# B  Ablation Studies

To determine the appropriate denoising steps and guidance scale, and the denoising strategy, we perform ablation studies by sampling the model with different parameters and testing the performance on the OpenTO benchmark mentioned in the experiments. Figure 4, shows the results of our ablation study on sampling parameters. To study the effects of denoising steps, we run both DDPM and DDIM denoising with a guidance scale of 2.0 with 5, 10, 20, and 40 denoising steps. We observe that DDIM sampling largely achieves better performance despite initially having a larger absolute VFE. Thus, we determine DDIM sampling to be more effective, especially for more efficient lower denoising step counts, crucial for computational efficiency. From here, we can clearly see in Figure 4 that DDIM sampling strikes the perfect balance of CE and VFE at 20 denoising steps, and any more comes with no benefits and, in fact, slightly worse performance. Finally, to determine the optimal guidance scale, we run DDIM sampling for guidance scales of 1, 2, 4, 6, 8, and 10 using the 20 denoising steps we found to be most effective. As seen in Figure 4, a guidance scale of 2 strikes a good balance between VFE and CE, which informs our final choice of DDIM denoising with 20 steps and a guidance scale of 2.

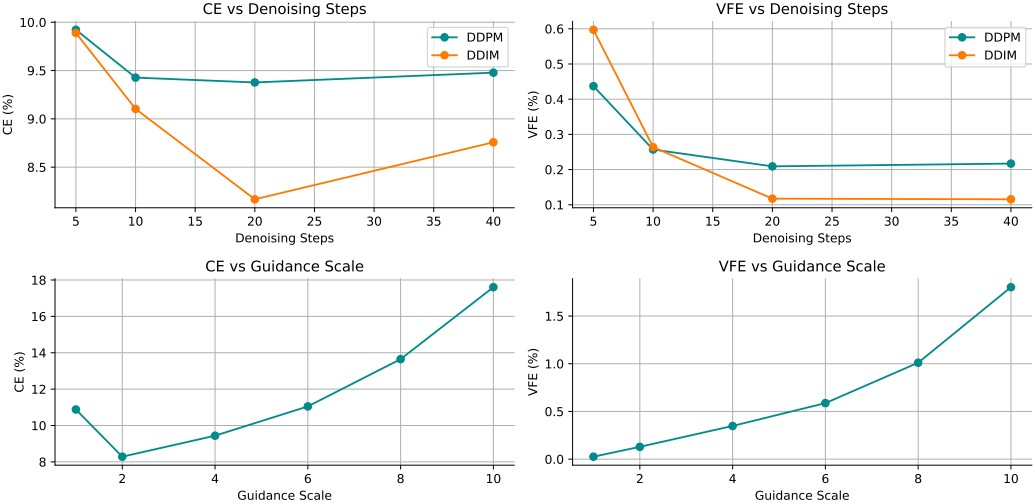

**Figure 4:** Top: Results of ablation studies on denoising steps. Observe that despite CE being lower in DDPM sampling, this is largely due to DDPM sampling using significantly more material, as evidenced by the much higher VFE values. Bottom: Ablation on the guidance scale shows that CE largely stays the same for the guidance scale of 1 and 2, while VFE is optimal at a guidance scale of 2.0.

# C   Out of Distribution Reconstruction

To further demonstrate potential downline adaptations of OAT as a foundational framework, we run reconstruction using our auto-encoder on completely out-of-distribution data, involving optimized topologies for heat sinks introduced by Bernard et al. [4]. This involves entirely different physics and optimization in comparison to the data we train OAT on. Despite this, we see an Intersection over Union (IoU) of 0.94 on these samples. The reconstruction quality is visualized in Figure 5. These results show how OAT, as a pre-trained foundation model, can easily be extended to different TO problems involving different physics and topologies, even without retraining the entire framework.

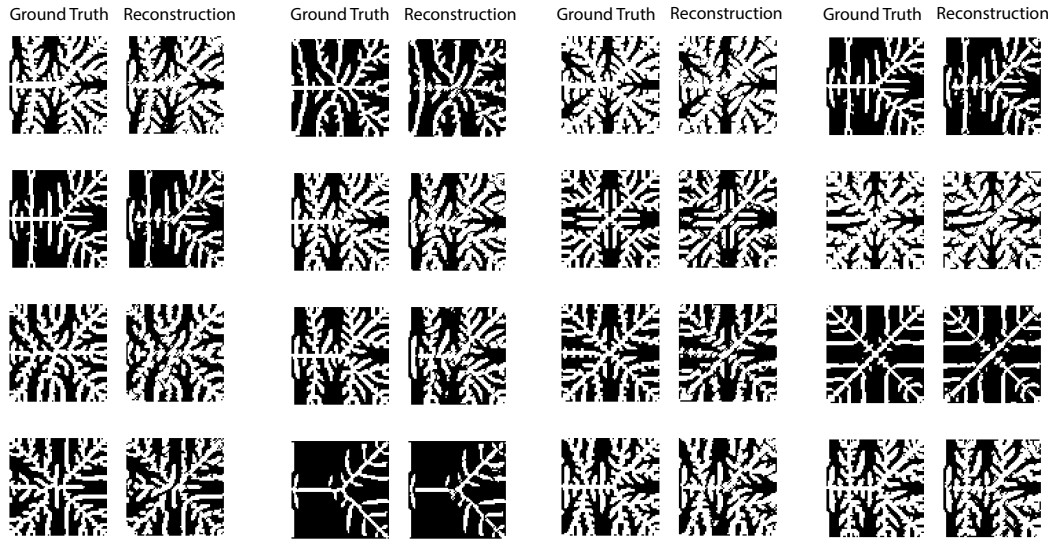

**Figure 5:** Heatsink optimized topologies reconstructed using OAT's auto-encoder without any additional training. This demonstrates OAT's zero-shot capability to extend to different physics and TO problems with relative ease.

# D  OpenTO Dataset

Here we describe the details of the OpenTO dataset and the data generation specifics, and at the end, provide visual examples of the topologies and boundary conditions in the problem.

The generation of each topology optimization problem within the OpenTO dataset follows a structured, randomized pipeline designed to produce a diverse set of problems. This diversity manifests in the domain discretization (resolution), aspect ratio, and the configuration of applied loads and boundary conditions. The overarching aim is to construct a comprehensive dataset suitable for foundation model training for TO.

## D.1  Domain Definition and Discretization

The first step involves defining the physical domain of the topology optimization problem. This is characterized by its resolution (total number of elements) and its aspect ratio.

1. **Target Element Count** ($EC$)**:** The total number of discrete elements in the design domain is sampled from a uniform distribution. Let $EC$ be the target element count, then

$$EC \sim \mathcal{U}[2^{12}, 2^{14}]$$

   This range ensures a variety of problem sizes, from $4096$ to $16384$ elements.

2. **Aspect Ratio** ($AR$)**:** The aspect ratio of the design domain is sampled from a standard log-normal distribution. Let $AR$ be the aspect ratio, then

$$AR \sim \text{LogNormal}(0, 1)$$

   where the parameters $(0, 1)$ represent the mean and standard deviation of the natural logarithm of the variable, respectively. This choice allows for a wide range of domain shapes, biased towards aspect ratios closer to unity but permitting more elongated or flattened domains. The bias towards square domains is mainly because these problems are more common and physically more stable, yielding fewer failed simulations in the optimization process.

3. **Problem Dimensions** ($N_x, N_y$)**:** Given the target element count $EC$ and the sampled aspect ratio $AR$, the dimensions of the problem domain, $N_x$ (number of elements in the x-direction) and $N_y$ (number of elements in the y-direction), are determined. Assuming $AR = N_x/N_y$, we have $N_x = AR \cdot N_y$. Since the total number of elements is $EC \approx N_x \cdot N_y$, we can substitute to get $EC \approx (AR \cdot N_y) \cdot N_y = AR \cdot N_y^2$. Thus,

$$N_y \approx \sqrt{\frac{EC}{AR}}$$

$$N_x \approx AR \cdot N_y$$

   The values for $N_x$ and $N_y$ are then rounded to the nearest integers, ensuring that their product $N_x \cdot N_y$ is close to the target $EC$. Furthermore, if this leads to an element count outside the intended distribution, one side is randomly adjusted to ensure this does not occur.

## D.2  Load and Constraint Specification

Once the domain is defined, loads and boundary constraints are applied. This involves determining the number of loads and constraints, their types, and their specific locations and orientations. Unlike prior datasets, which hand-select a finite number of boundary conditions and apply a single load on the boundary, OpenTO samples boundary conditions randomly and applies multiple random loads of different kinds. Furthermore, unlike prior works, which limit forces and boundary conditions to the edges of the domain, we sample internal boundary conditions and loads for full generality. OpenTO also includes loads, which we refer to as distributed loads, which are typically characterized formally as Neumann boundary conditions when loads are a result of stress/pressure applied at the boundaries, and body forces when an internal load is applied to part of the domain interior (often seen in electromagnetic forces or gravitational body forces). Below are details of how loads and boundary conditions are sampled in OpenTO:

1. **Number of Loads** ($NL$)**:** The number of applied loads is sampled from a geometric distribution with parameter $p = 0.3$, shifted by $+1$ to ensure at least one load. Let $NL$ be the number of loads, then

$$NL \sim \text{Geom}(0.3) + 1$$

2. **Number of Constraints ($NC$):** The number of boundary constraints (fixed degrees of freedom) is sampled from a geometric distribution with parameter $p = 0.2$, shifted by $+2$ to ensure at least two constraints. Let $NC$ be the number of constraints, then

$$NC \sim \text{Geom}(0.2) + 2$$

This ensures a baseline level of constraint necessary for a valid mechanical problem. If fewer constraints are present, the FEA problem (Appendix A) will become singular and thus not solvable.

3. **Load and Constraint Types and Placement:** For each of the $NL$ loads and $NC$ constraints, a type is selected from a predefined discrete probability distribution. The available types and their selection probabilities are:

   - Internal point force (single force in the interior of the domain): $50\%$
   - Edge point (a point on one of the four edges, not a corner): $10\%$
   - Corner point (one of the four corners): $10\%$
   - Distributed load/constraint (equal magnitude applied across many nodes of a line segment on one edge) on a partial edge: $10\%$
   - Distributed load/constraint on a full edge: $10\%$
   - Internally distributed load/constraint (distributed on a line or area of a random ellipse): $10\%$

   The specific placement of a load or constraint is then determined randomly based on the selected type. For example:

   - *Internal point:* A random coordinate $(x, y)$ within the domain interior.
   - *Edge point:* An edge (top, bottom, left, or right) is chosen uniformly at random, and then a random position along that edge is selected.
   - *Corner point:* One of the four domain corners $((0,0), (N_x, 0), (0, N_y), (N_x, N_y)$ in elemental coordinates, or corresponding nodal coordinates) is chosen uniformly at random.
   - *Distributed on partial edge:* An edge is chosen, and then a random sub-segment of that edge is selected.
   - *Distributed on full edge:* An edge is chosen.
   - *Internally distributed:* A region (e.g., a rectangular or circular patch or line) is randomly defined within the interior of the domain. The specific shapes and parameters for these internal distributions are further randomized. For comprehensive details on the implementation of these variations, readers are directed to the project's codebase.

   Loads are typically defined by a vector (e.g., magnitude and direction), which can also be randomized.

4. **Constraint Direction:** Each of the $NC$ constraints is assigned a direction of application. The degrees of freedom (DOFs) to be constrained are chosen based on the following probabilities:

   - Constraint in the lateral (e.g., x-direction) direction only: $30\%$
   - Constraint in the vertical (e.g., y-direction) direction only: $30\%$
   - Constraint in both lateral and vertical directions: $40\%$

## D.3 Problem Validation and Iteration

A critical final step in the generation pipeline is the validation of the constructed problem.

1. **Validity Check:** The problem, now fully defined by its domain, loads, and constraints, is assessed to ensure it is:

   - **Fully constrained:** The applied boundary conditions must be sufficient to prevent rigid body motion (translation and rotation) of the structure under the applied loads. This is typically checked by ensuring the global stiffness matrix is nonsingular.
   - **Not trivially solved:** The problem should not be over-constrained to the point of having an obvious or degenerate solution. For instance, applying a load and a fixed constraint at the exact same degree of freedom might lead to issues or trivial (zero vector) forcing terms.

2. **Regeneration on Failure:** If the generated problem fails the validity check (e.g., due to coincident load and constraint application points, insufficient constraints, or other problematic configurations), the entire problem instance is discarded. The generation process,

starting from the sampling of $EC$, is then re-initiated to produce a new, statistically independent problem candidate. This iterative loop continues until a valid problem instance is successfully generated.

This rigorous, multi-stage randomized procedure ensures the creation of a diverse and challenging set of topology optimization problems, forming the basis of the OpenTO dataset.

## D.4 Algorithm for Data Generation

Below is a summary of the data generation process in algorithmic form.

---

**Algorithm 1** OpenTO Problem Instance Generation

---

1: **Initialize:** ProblemIsValid ← false
2: **while** ProblemIsValid is false **do**
3:     *// Step 1: Domain Definition and Discretization*
4:     Sample target element count $EC \sim \mathcal{U}[2^{12}, 2^{14}]$
5:     Sample aspect ratio $AR \sim \text{LogNormal}(0, 1)$
6:     Calculate $N_{y,float} \leftarrow \sqrt{EC/AR}$
7:     Calculate $N_{x,float} \leftarrow AR \cdot N_{y,float}$
8:     Set $N_y \leftarrow \text{round}(N_{y,float})$
9:     Set $N_x \leftarrow \text{round}(N_{x,float})$
10:     **if** $N_x < 1$ **or** $N_y < 1$ **then**
11:         **continue** *// Restart generation if dimensions are invalid*
12:     **end if**
13:     *// Step 2: Load and Constraint Specification*
14:     Sample number of loads $NL \sim \text{Geom}(0.3) + 1$
15:     Sample number of constraints $NC \sim \text{Geom}(0.2) + 2$
16:     Initialize empty lists: LoadsList, ConstraintsList
17:     **for** $i = 1$ **to** $NL$ **do**
18:         Sample load type $L_{type}$ from
        {InternalPt (0.5), EdgePt (0.1), CornerPt (0.1), PartialEdge (0.1), FullEdge (0.1), InternalDist (0.1)}
19:         Determine load placement $L_{place}$ based on $L_{type}$ and domain $(N_x, N_y)$ (randomized selection)
20:         Determine load magnitude and direction $L_{vec}$ (randomized)
21:         Add $(L_{type}, L_{place}, L_{vec})$ to LoadsList
22:     **end for**
23:     **for** $j = 1$ **to** $NC$ **do**
24:         Sample constraint type $C_{type}$ from {InternalPt (0.5), EdgePt (0.1), CornerPt (0.1), PartialEdge (0.1), FullEdge (0.1), InternalDist (0.1)}
25:         Determine constraint placement $C_{place}$ based on $C_{type}$ and domain $(N_x, N_y)$ (randomized selection)
26:         Sample constraint direction $C_{dir}$ from {Lateral (0.3), Vertical (0.3), Both (0.4)}
27:         Add $(C_{type}, C_{place}, C_{dir})$ to ConstraintsList
28:     **end for**
29:     *// Step 3: Problem Validation*
30:     Perform check for sufficient constraints (prevent rigid body motion).
31:     Perform check for non-trivial solution (e.g., avoid coincident loads/constraints at same DOF if problematic).
32:     **if** problem is fully constrained AND not trivially solved **then**
33:         ProblemIsValid ← true
34:     **else**
35:         ProblemIsValid ← false *// Problem discarded, loop will reiterate*
36:     **end if**
37: **end while**
38: **Output:** Generated problem instance (Domain: $N_x, N_y$; Loads: LoadsList; Constraints: ConstraintsList) =0

---

# E   Further Visualization of Results

Here we visualize more samples from OAT and NITO on the fully general OpenTO benchmark. The figures below clearly highlight how NITO fails to generate high-quality samples and generates largely invalid, blurry topologies that are not mostly not binary as ideally would be in TO. NOTE: Samples with CE higher than 100% are considered failed and do not enter CE computation in the reported results.

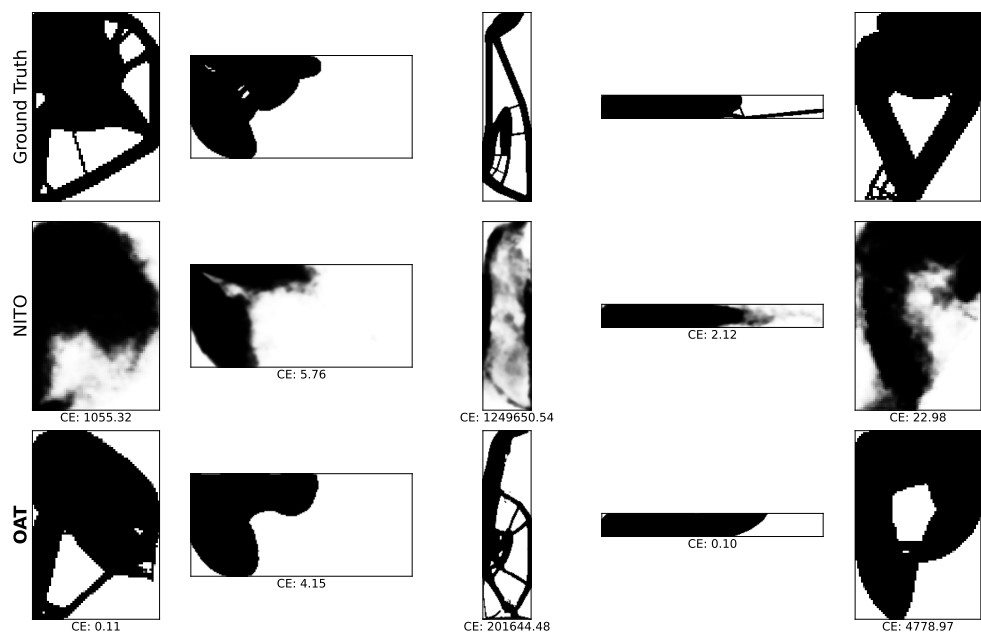

**Figure 6:** Samples generated on the OpenTO benchmark by NITO and OAT. Compliance error for each generated sample is shown underneath.

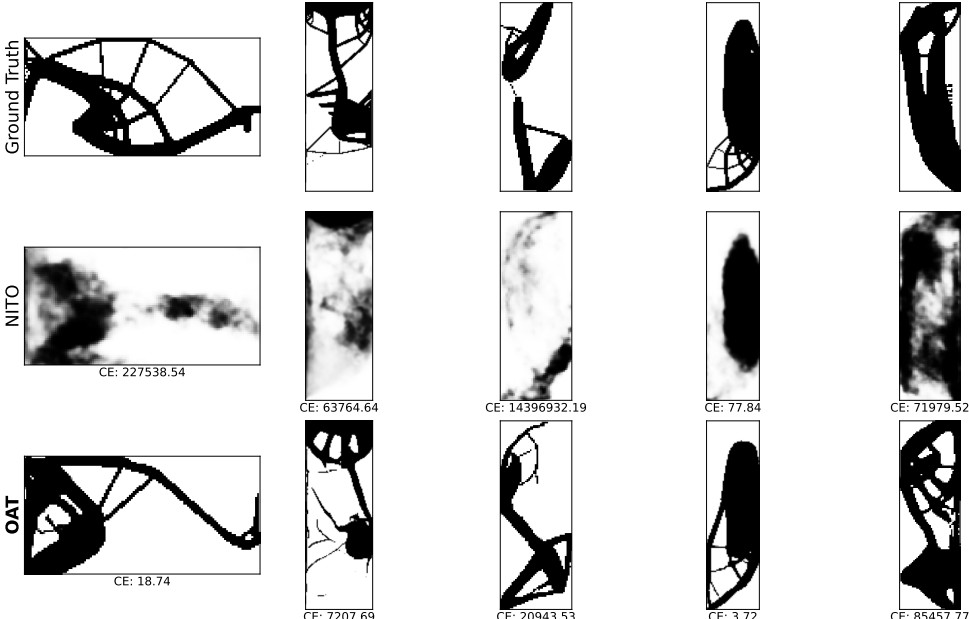

**Figure 7:** Samples generated on the OpenTO benchmark by NITO and OAT. Compliance error for each generated sample is shown underneath.

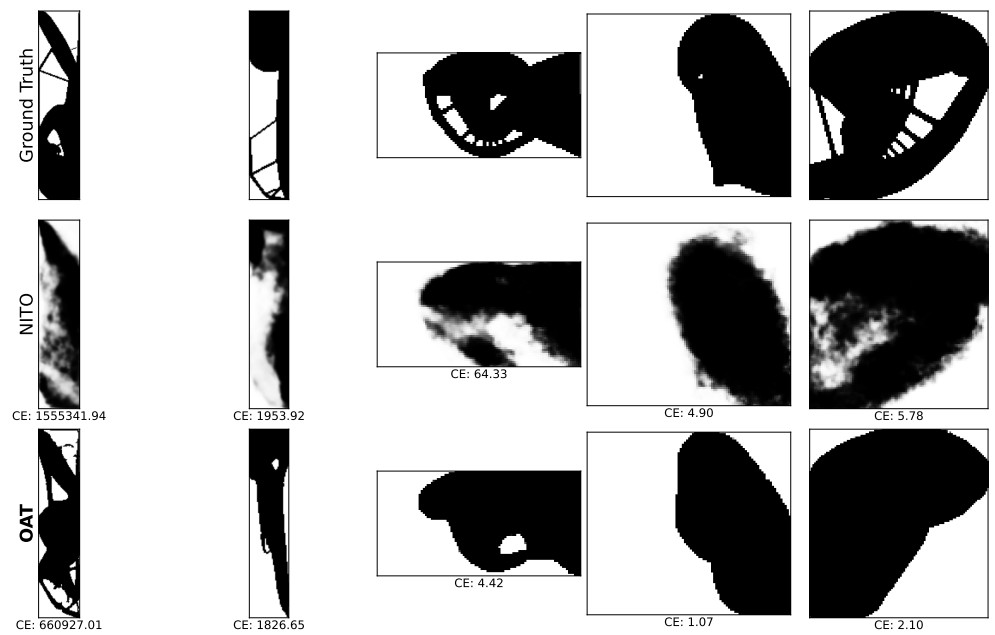

**Figure 8:** Samples generated on the OpenTO benchmark by NITO and OAT. Compliance error for each generated sample is shown underneath.

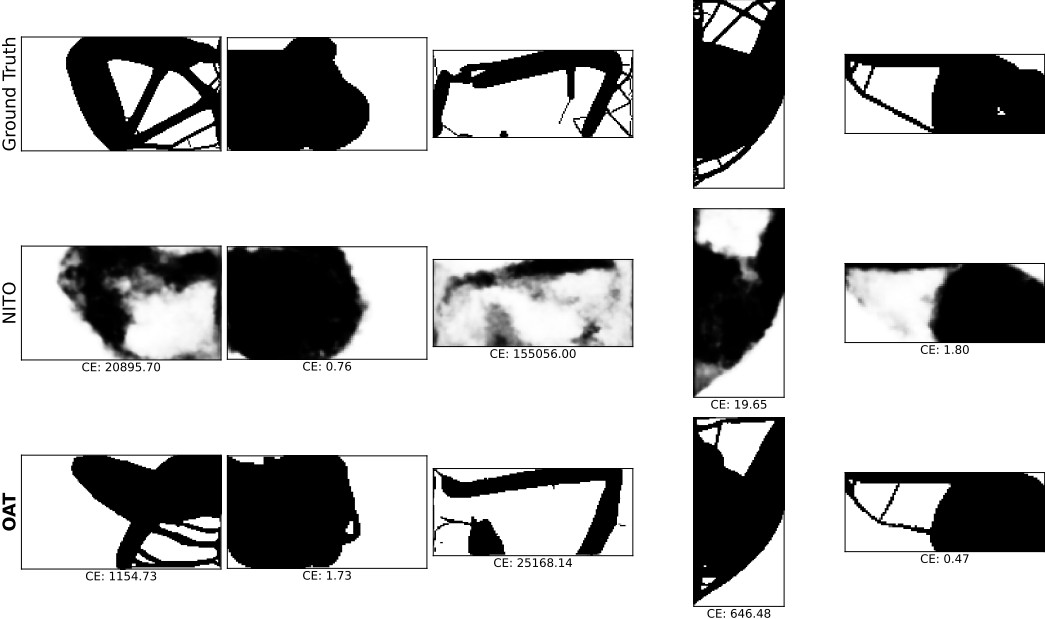

**Figure 9:** Samples generated on the OpenTO benchmark by NITO and OAT. Compliance error for each generated sample is shown underneath.

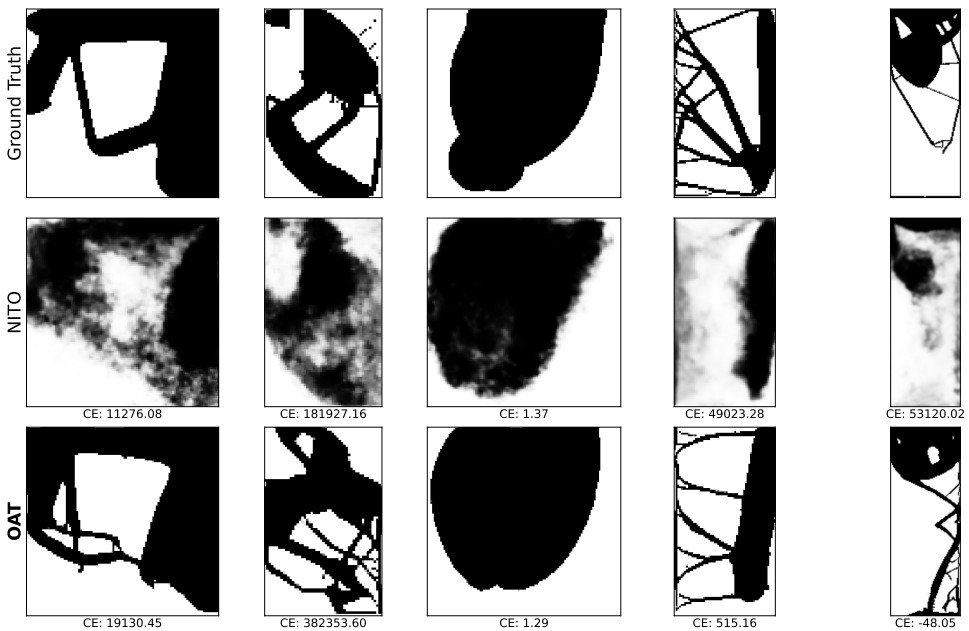

**Figure 10:** Samples generated on the OpenTO benchmark by NITO and OAT. Compliance error for each generated sample is shown underneath.

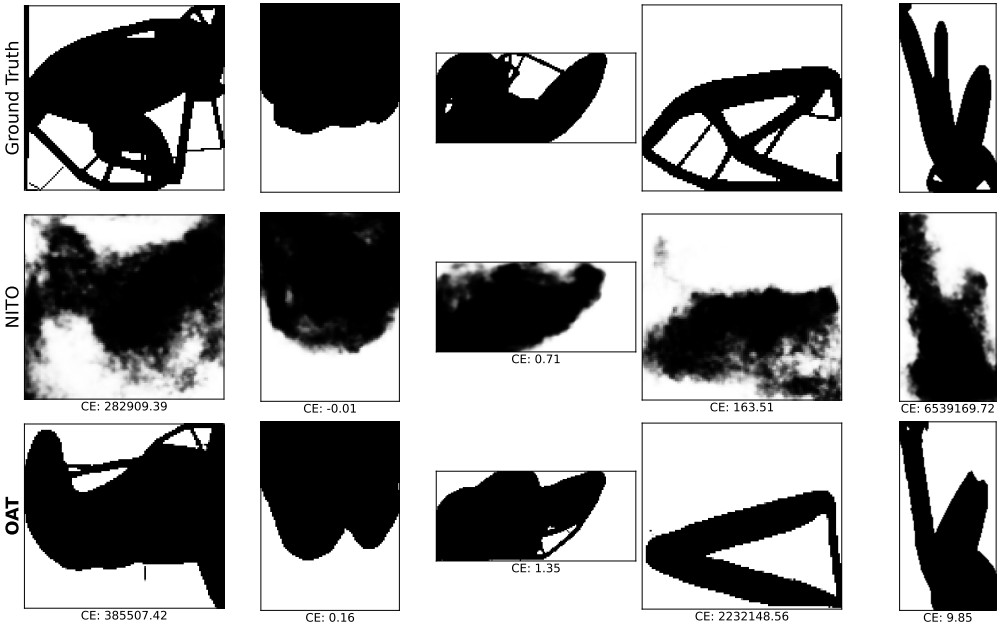

**Figure 11:** Samples generated on the OpenTO benchmark by NITO and OAT. Compliance error for each generated sample is shown underneath.

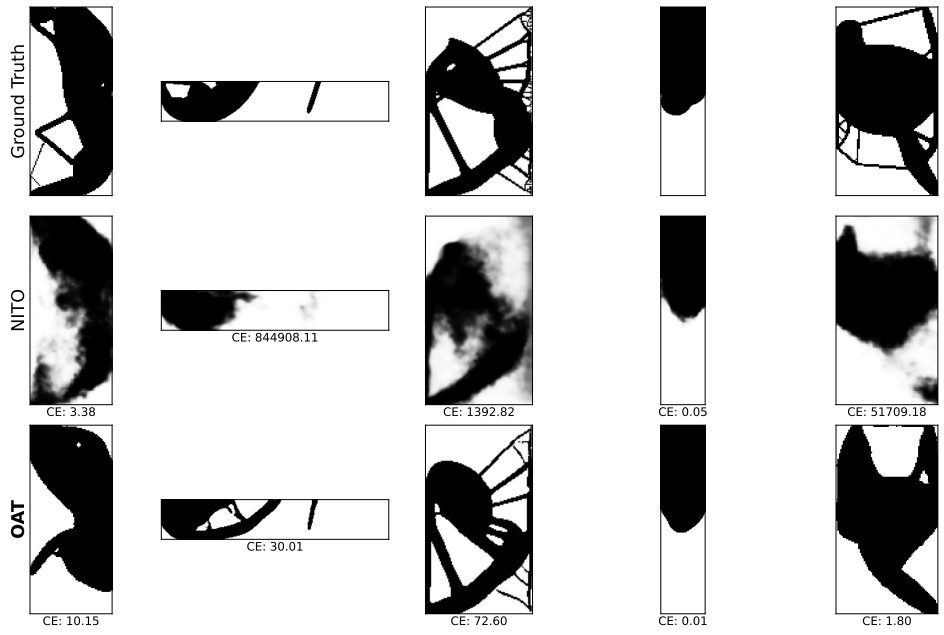

**Figure 12:** Samples generated on the OpenTO benchmark by NITO and OAT. Compliance error for each generated sample is shown underneath.

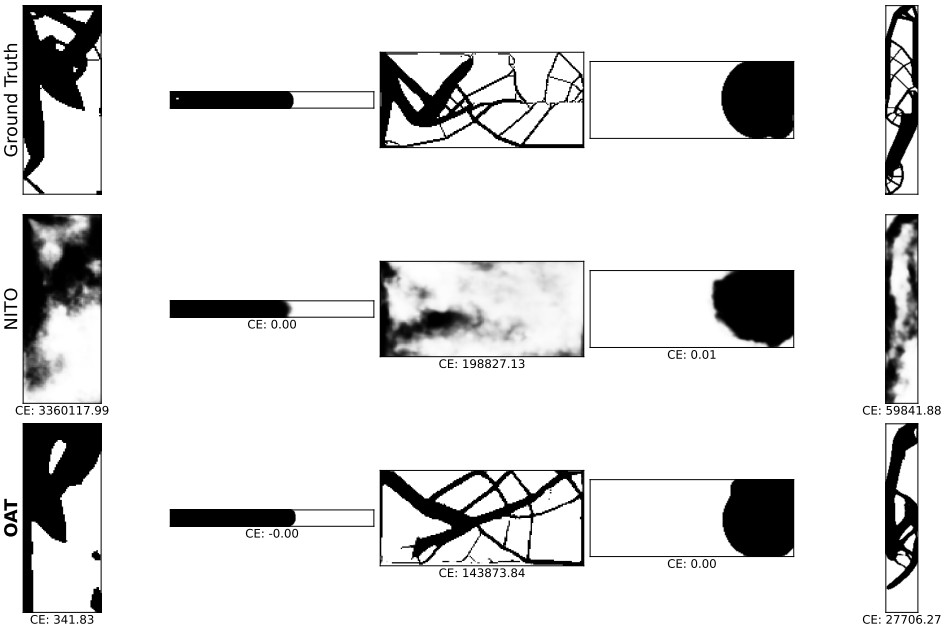

**Figure 13:** Samples generated on the OpenTO benchmark by NITO and OAT. Compliance error for each generated sample is shown underneath.

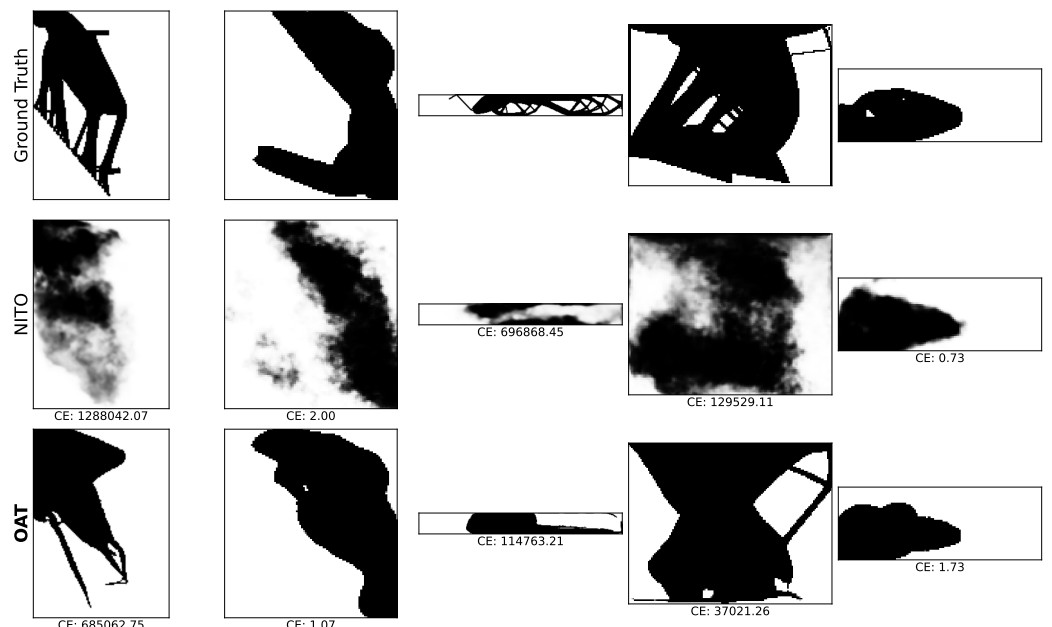

**Figure 14:** Samples generated on the OpenTO benchmark by NITO and OAT. Compliance error for each generated sample is shown underneath.

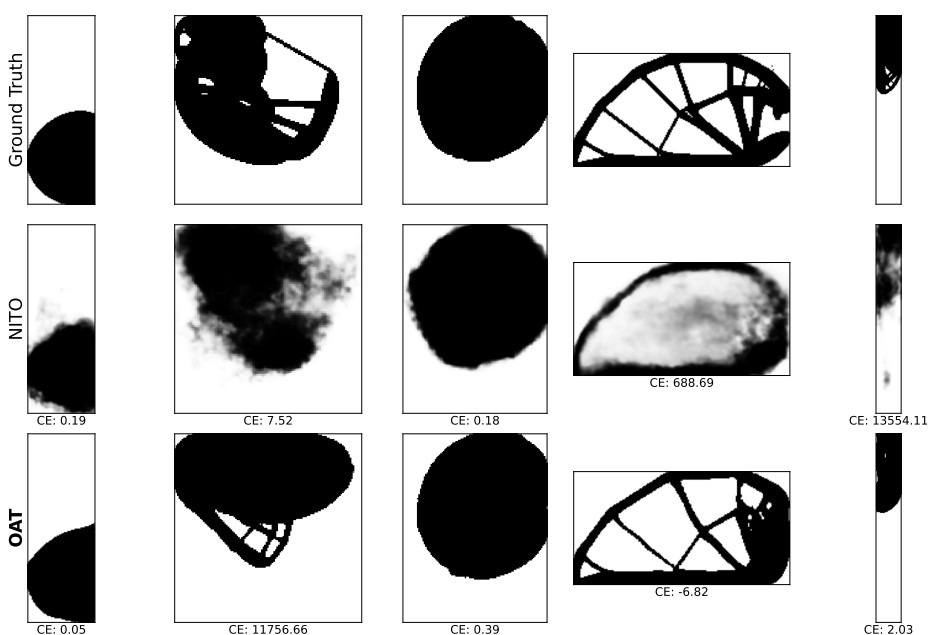

**Figure 15:** Samples generated on the OpenTO benchmark by NITO and OAT. Compliance error for each generated sample is shown underneath.

## F  Implementation Details

In this section, we describe the full details of the training and architecture of OAT. First, we discuss the autoencoder training, then we will go over the LDM training.

### F.1  Autoencoder Architecture

The autoencoder comprises an encoder, a decoder, and a neural field renderer.

**Encoder**  The encoder maps an input topology $T$ to a latent representation $z = E(T)$.

- **Input Processing:** Topologies are padded and resized to a fixed $256 \times 256$ resolution with 1 input channel.
- **Core Structure:** A Convolutional Neural Network (CNN) with an initial convolution is followed by three downsampling levels, each employing ResNet-style blocks.
- **Middle Section:** Features are further processed by a middle section containing additional ResNet-style blocks and an attention mechanism.
- **Output:** Final convolutional layers produce a 1-channel latent tensor of size 64 x 64.

**Decoder**  The decoder reconstructs a feature tensor $\phi = D(z)$ from the latent $z$.

- **Input:** The latent tensor $z$ is processed by initial convolutional layers.
- **Core Structure:** Symmetrical to the encoder, it features a middle section with ResNet-style blocks and an attention mechanism, followed by three upsampling levels composed of ResNet-style blocks.
- **Output:** A fixed-resolution feature tensor (256 x 256) $\phi$ with 128 channels.

**Neural Field Renderer**  The renderer reconstructs the topology $\tilde{T} = R(\phi, c, s)$ from $\phi$, pixel coordinates $c$, and cell sizes $s$.

- **Architecture:** A convolutional renderer, inspired by Convolutional Local Image Functions (CLIF), is employed. It processes the decoded features, coordinates, and cell sizes, with optional positional encoding for the latter two.
- **Network:** The renderer consists of a sequence of convolutional and ResNet-style blocks, terminating in an output convolution with a Tanh activation function.
- **Scalability:** Training utilizes patch sampling. High-resolution inference employs stenciled, overlapping tiles.

Full details of the architecture can be found in our publicly available code.

### F.2  Autoencoder Training

The autoencoder is trained on the OpenTO dataset (2.2 million pre-training samples).

- **Dataset and Preprocessing:**
  - Input topologies are resized to $256 \times 256$ for the encoder.
  - The renderer is trained on $64 \times 64$ patches sampled from topologies, along with their coordinates and cell sizes. Training can also be performed on full images (full grid of coordinates and cell sizes for any given topology).
- **Batch Size and Epochs:**
  - Batch size: 128.
  - Training epochs: 50.
- **Distributed Training and Precision:**
  - Training utilizes Distributed Data Parallel (DDP) on 4 H100 GPUs.
  - Automatic Pytorch mixed precision training is employed.
- **Optimizer and Learning Rate Schedule:**
  - AdamW optimizer is used for training.
  - A Cosine schedule is used for learning rate with 200 steps of warmup, linearly increasing learning rate from 0 to $10^{-4}$, then gradually reducing it to $10^{-5}$ during training.

## F.3 Latent Diffusion Architecture

The latent diffusion model operates on the latent encodings of the autoencoder, which have a 64 x 64 resolution with 1 channel. The architecture of the model follows a UNet architecture, which can be seen in full in our publicly available code. For conditioning of the LDM model, we use a problem encoder architecture similar to what we described in the main body of the paper:

**Conditioning Mechanism**

- **Time Embedding:** Standard timestep embedding is used to encode the diffusion timestep $t$.
- **Problem Embedding ($P$):** The TO problem definition $\hat{P}$ (boundary conditions, forces, volume fraction, aspect ratio, cell size) is encoded into a fixed-size problem embedding $P$ using a dedicated `ProblemEncoder`.
  - Boundary conditions ($S_{boundary}$) and forces ($S_{force}$), represented as point clouds, are processed by separate BPOM modules (MLPs with mean/max/min pooling of point features concatenated).
  - Scalar and low-dimensional conditions (volume fraction $VF$, cell size $s$, aspect ratio $a$) are processed by individual MLPs.
  - The concatenated embeddings are passed through a final MLP to produce $P$.
- **Combined Embedding:** The problem embedding $P$ is projected to the same dimension as the time embedding and added to it. This combined embedding conditions the ResNet and Attention blocks within the U-Net.

For full details of the architecture, please refer to the code we provide.

## F.4 latent Diffusion Training

The LDM is trained on latent codes $z$ obtained from the pre-trained autoencoder, paired with their corresponding TO problem specifications $\hat{P}$.

- **Dataset:** Consists of latent tensors and their associated problem definitions (forces, boundary conditions, volume fraction, etc.). The dataset loader handles stochastic dropping of conditions for classifier-free guidance.
- **Diffusion Process:**
  - A DDPM noise schedule is used with a 'velocity' target, and a cosine noise schedule for training.
  - For inference, a DDIM noise scheduler is employed for faster sampling.
- **Training Objective:** The model is trained to predict the velocity $v$ of the diffusion process, conditioned on the problem embedding $P$. The objective is to minimize the mean squared error between the true and predicted velocity, as described in Equation 6 of the main paper:

$$\mathcal{L}_{LDM} = \mathbb{E}_{z \sim E(T), t, \epsilon \sim \mathcal{N}(0,I)}[||v - v_\theta(z_t, t|P)||^2],$$

  where $v = \sqrt{\overline{\alpha_t}}\epsilon - \sqrt{1 - \overline{\alpha_t}}z$.
- **Classifier-Free Guidance:** Enabled by randomly setting the problem conditioning $P$ to a null embedding during training with a specified probability, specifically, we hide all of $P$ 50% of the time and hide boundary conditions and forces each with a probability of 25%, separate from the full 59% hiding.
- **Optimizer and Learning Rate:**
  - AdamW optimizer.
  - Initial learning rate: $10^{-4}$, with 200 warmup steps same as for the autoencoder.
  - A cosine learning rate scheduler anneals the learning rate to a final value of $10^{-5}$.
- **Batch Size and Epochs:**
  - Batch size: 64.
  - Training epochs: 50.
- **Distributed Training and Precision:**
  - Training utilizes Distributed Data Parallel (DDP) on 2 H100 GPUs.
  - Mixed precision training is employed.

## F.5 Computational Resources

All experiments and training were conducted using 4 H100 GPUs. Autoencoder training takes 2 days to complete on our implementation, and the diffusion model takes 4 days to train. We had multiple training runs, but we do not have an exact estimate of the time needed. Inference time is indicated for the RTX 4090 GPU in our model, which was used for inference experiments.

## F.6 Dataset Visualization

Here we provide a few examples of the OpenTO dataset we generate. We visualize forces and boundary conditions, and it can be seen that OpenTO contains samples that include non-repeating complex problem definitions with internal, edge, distributed, and point boundary conditions and forces.

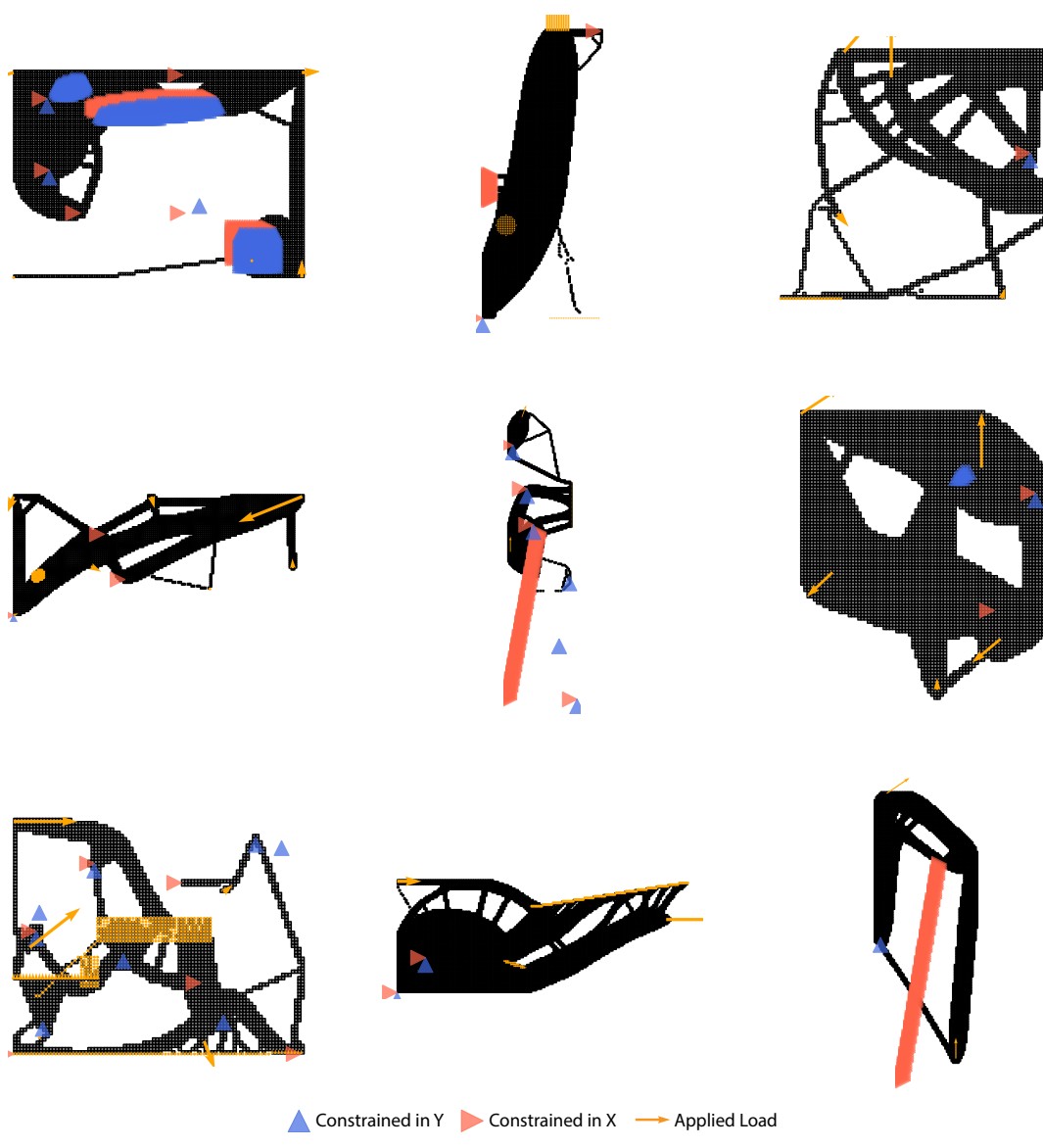

**Figure 16:** Samples from OpenTO Dataset.

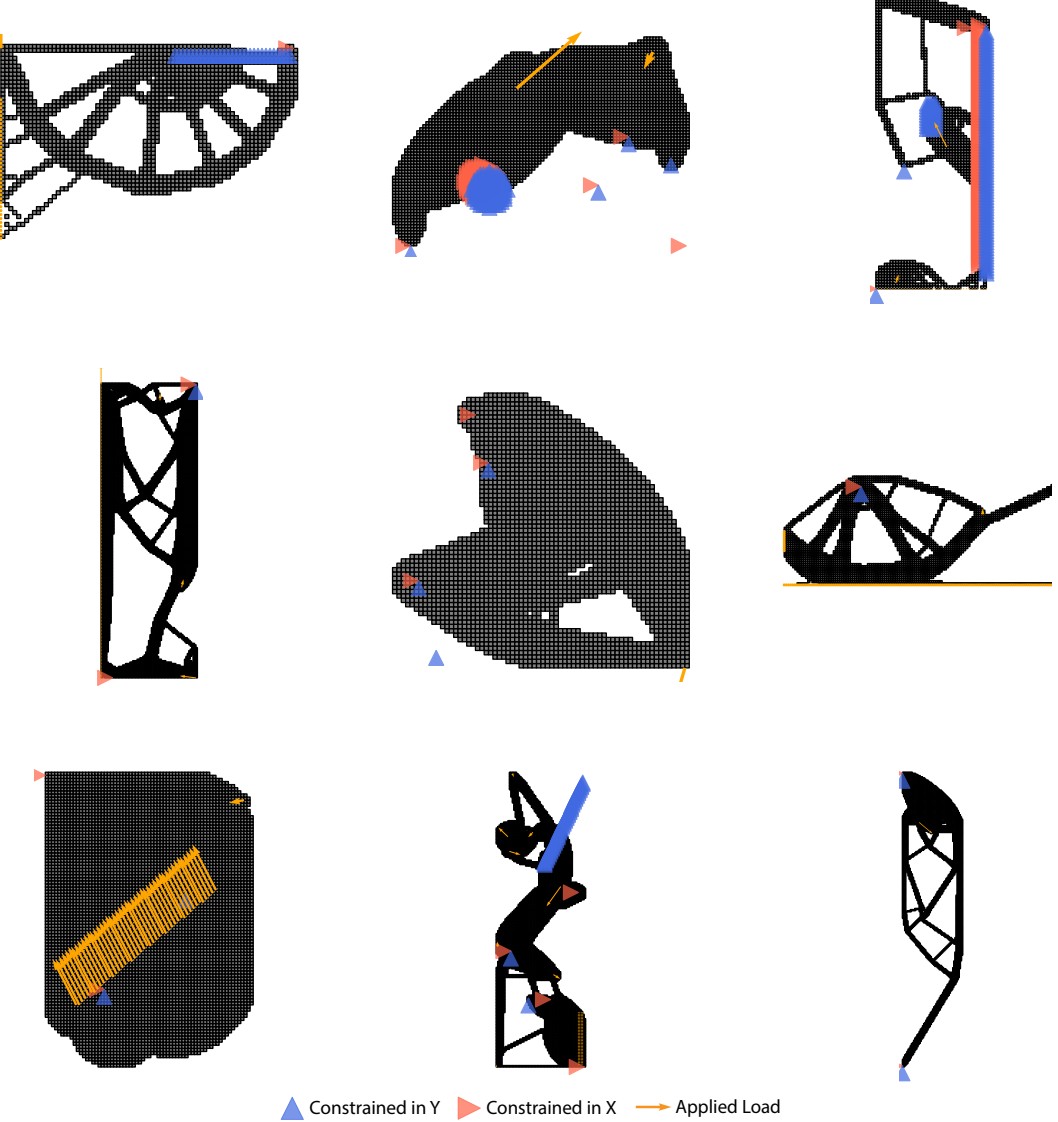

**Figure 17:** Samples from OpenTO Dataset.

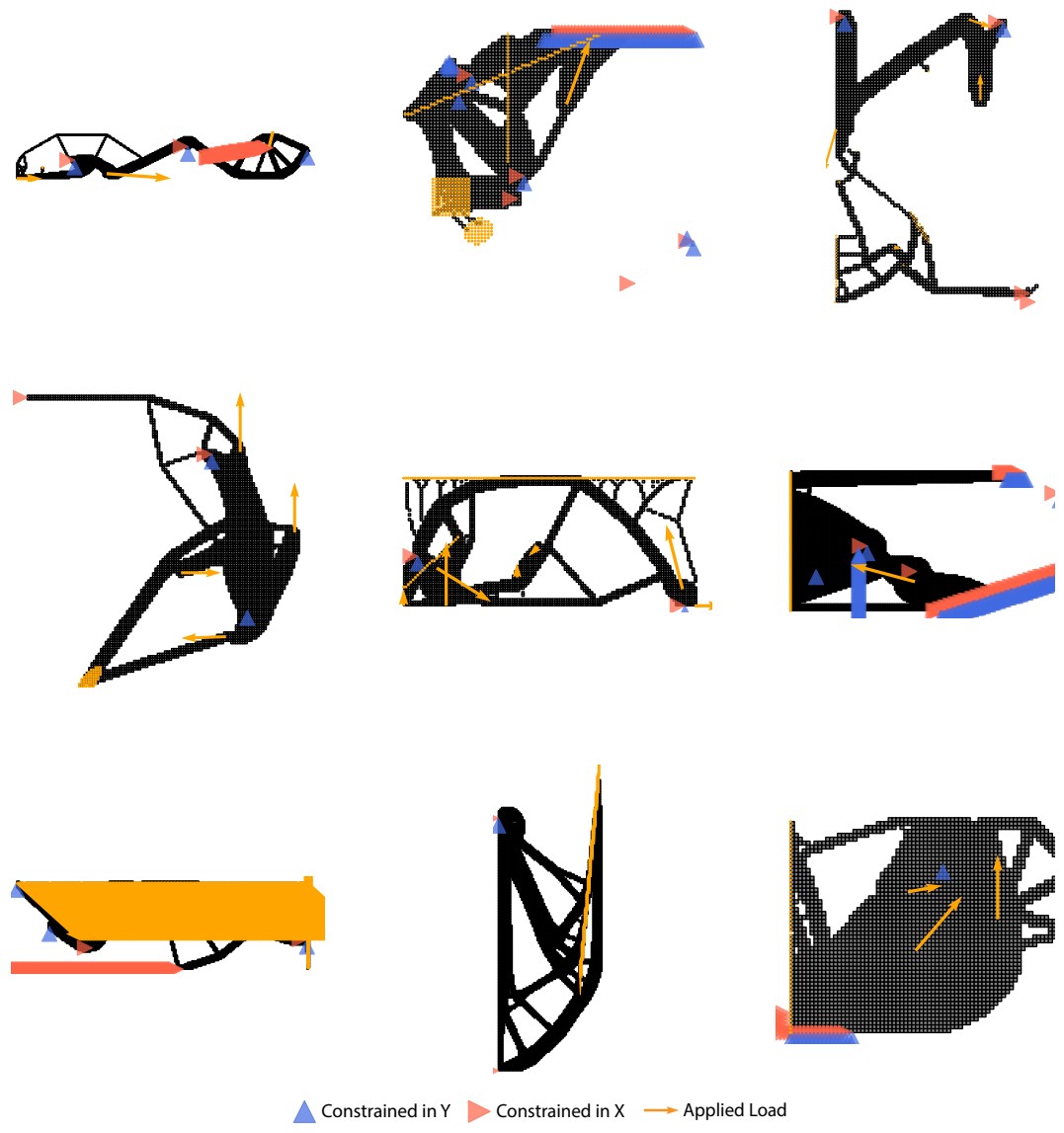

**Figure 18:** Samples from OpenTO Dataset.

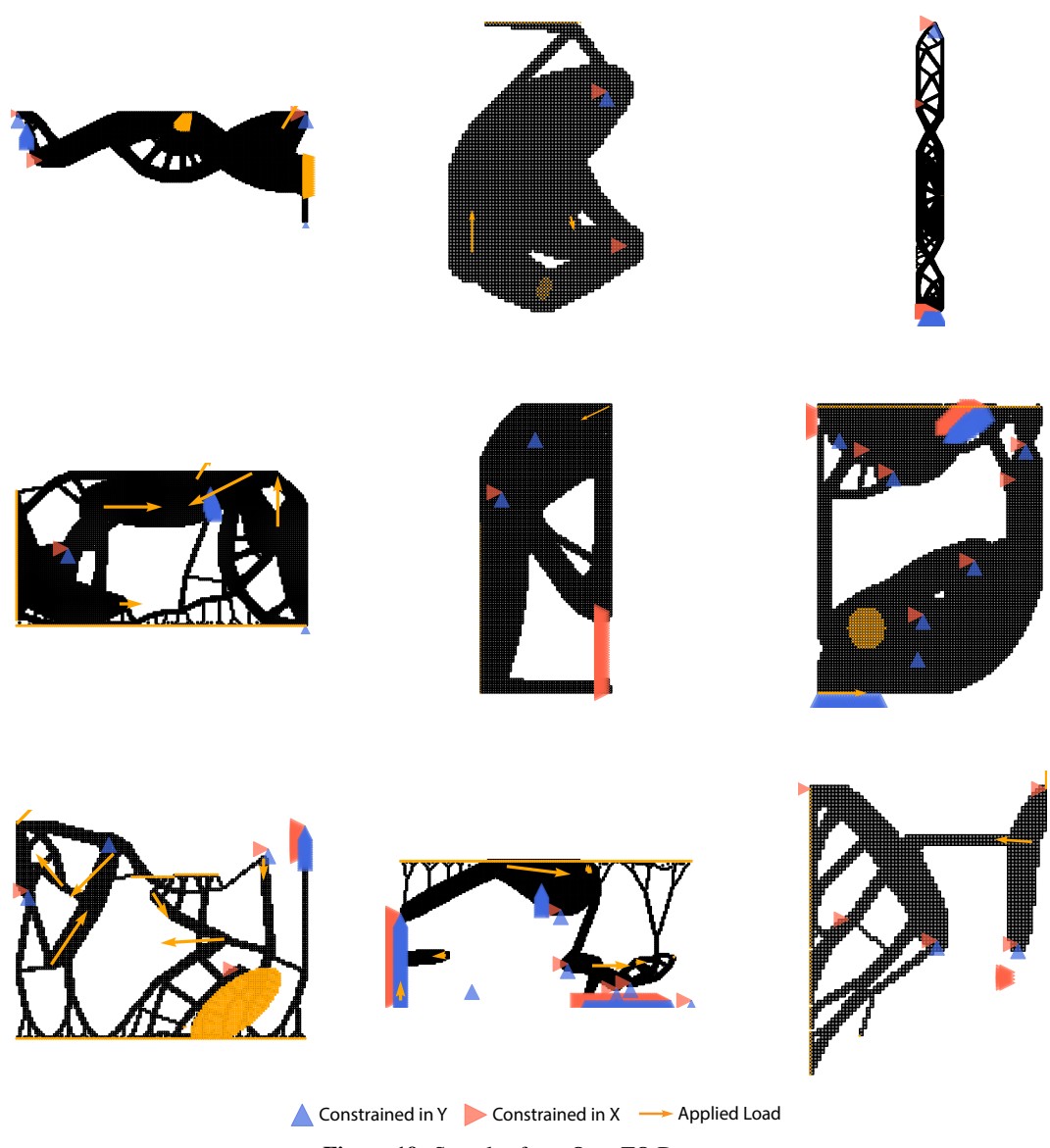

**Figure 19:** Samples from OpenTO Dataset.

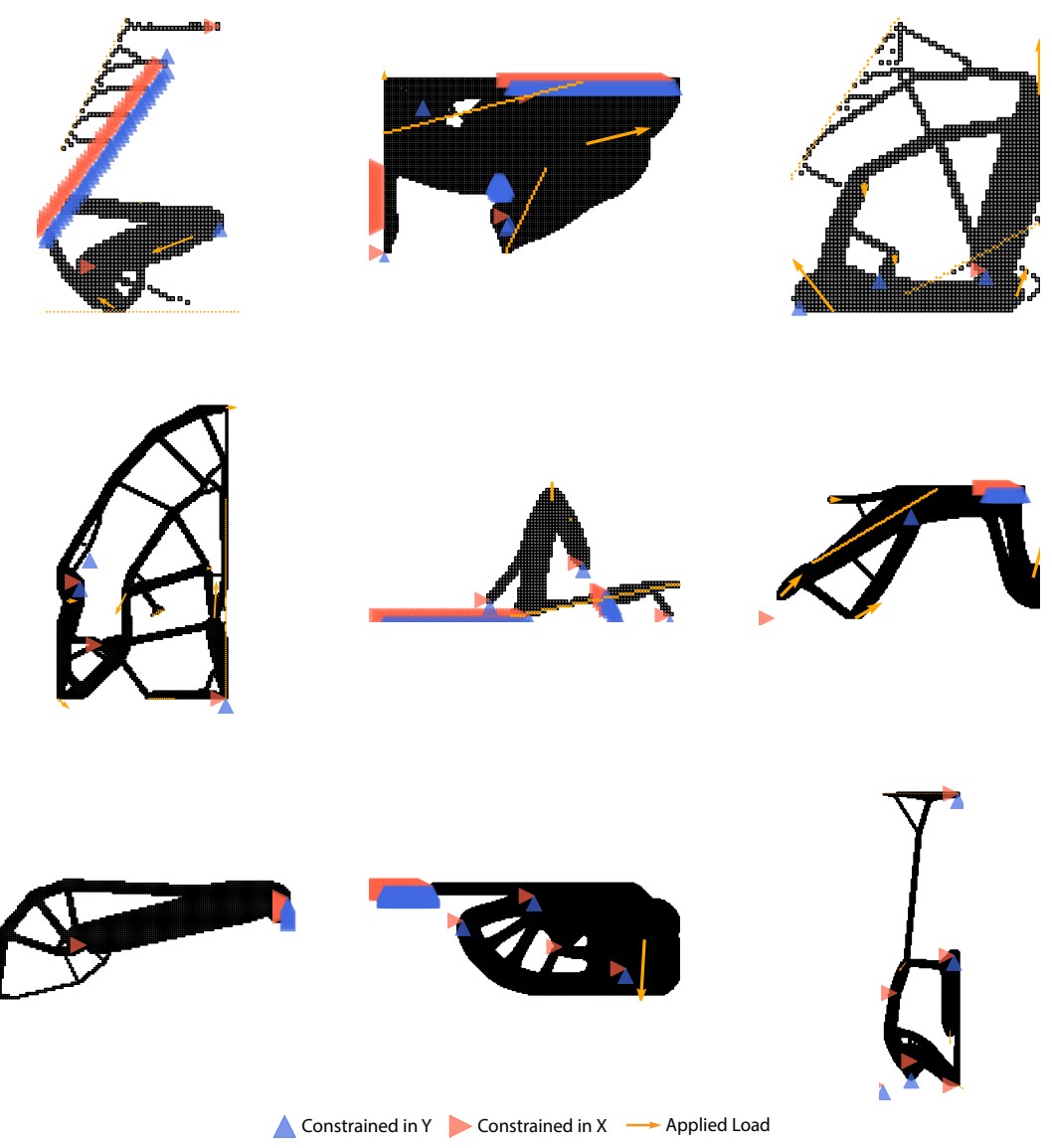

**Figure 20:** Samples from OpenTO Dataset.

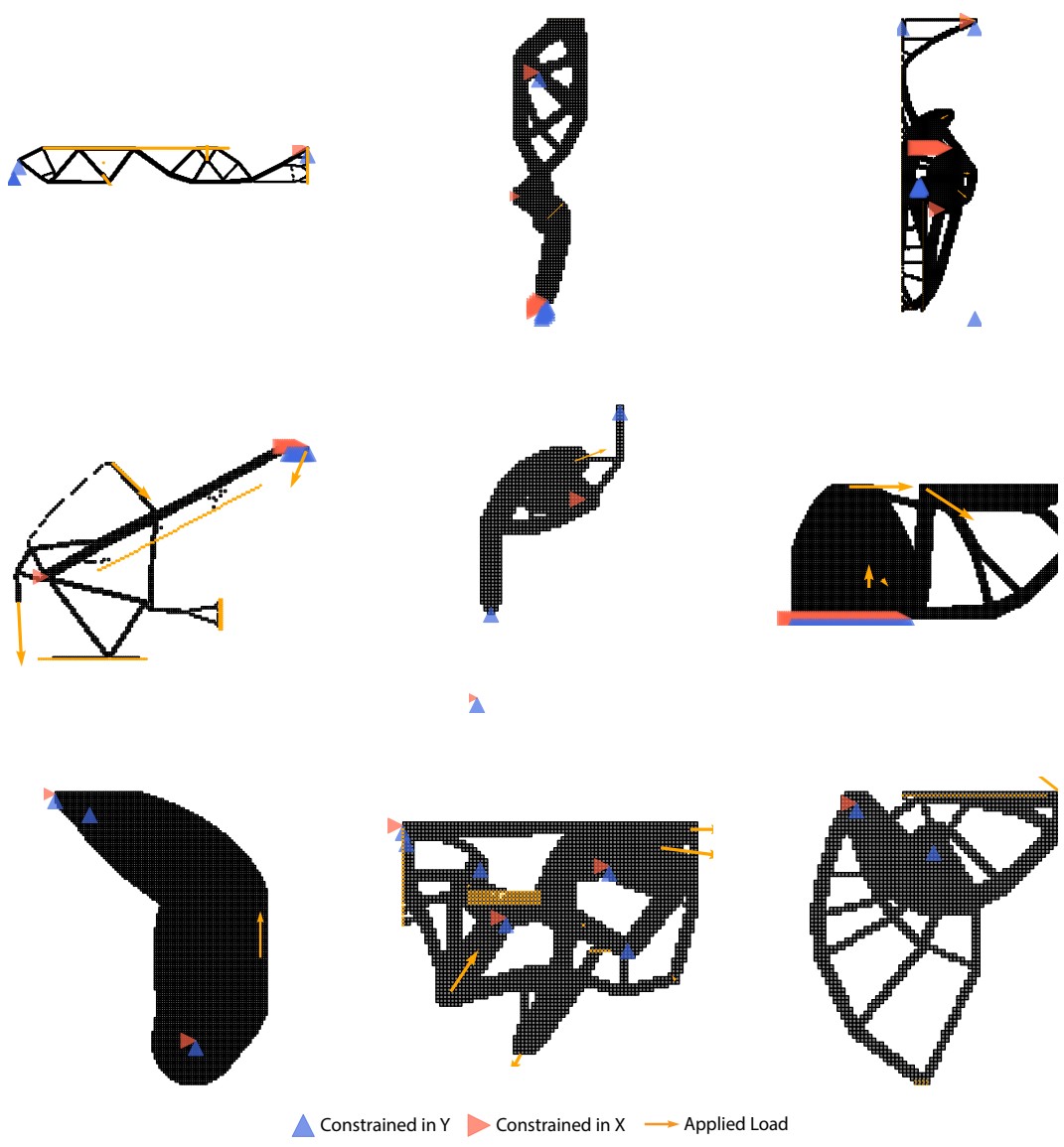

**Figure 21:** Samples from OpenTO Dataset.

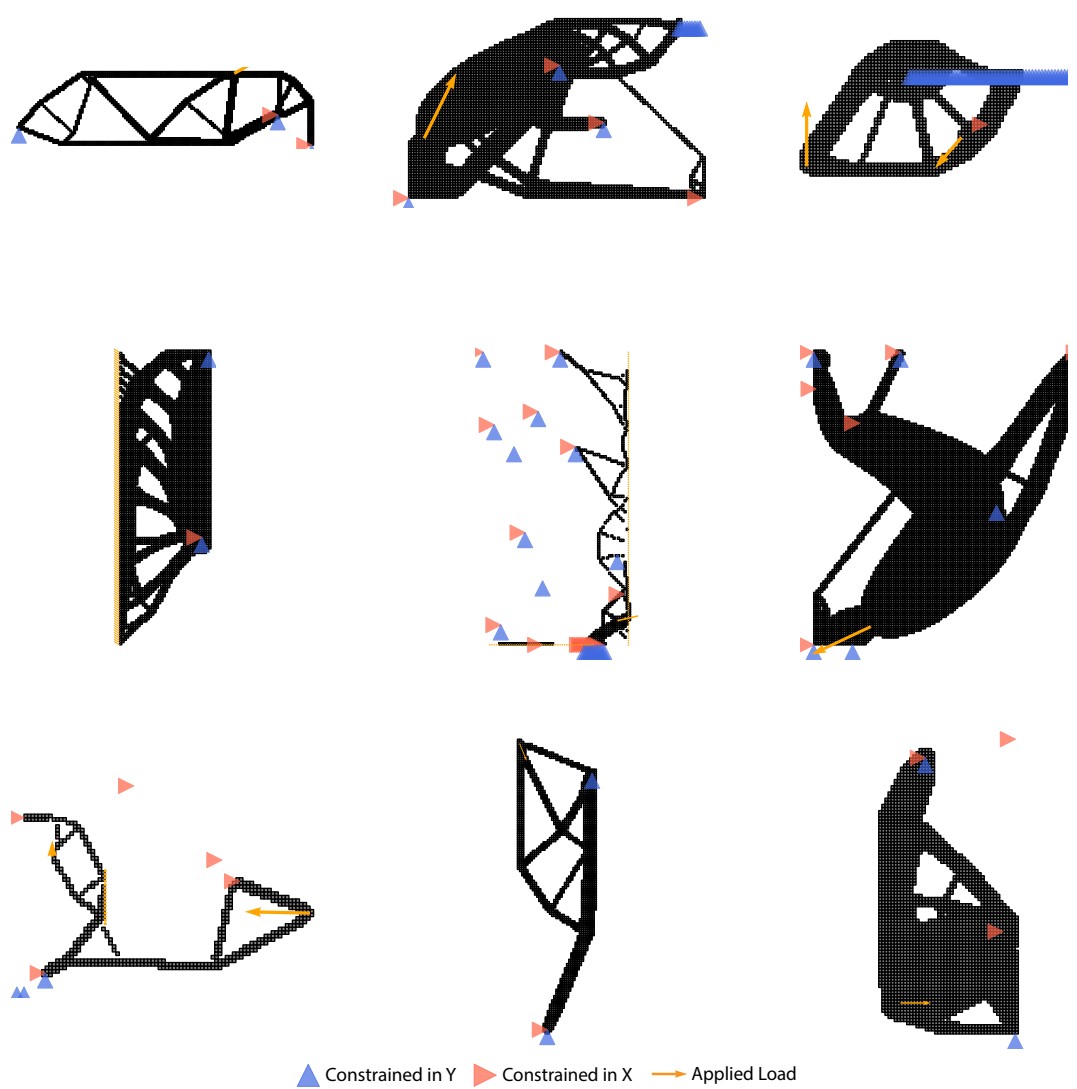

**Figure 22:** Samples from OpenTO Dataset.

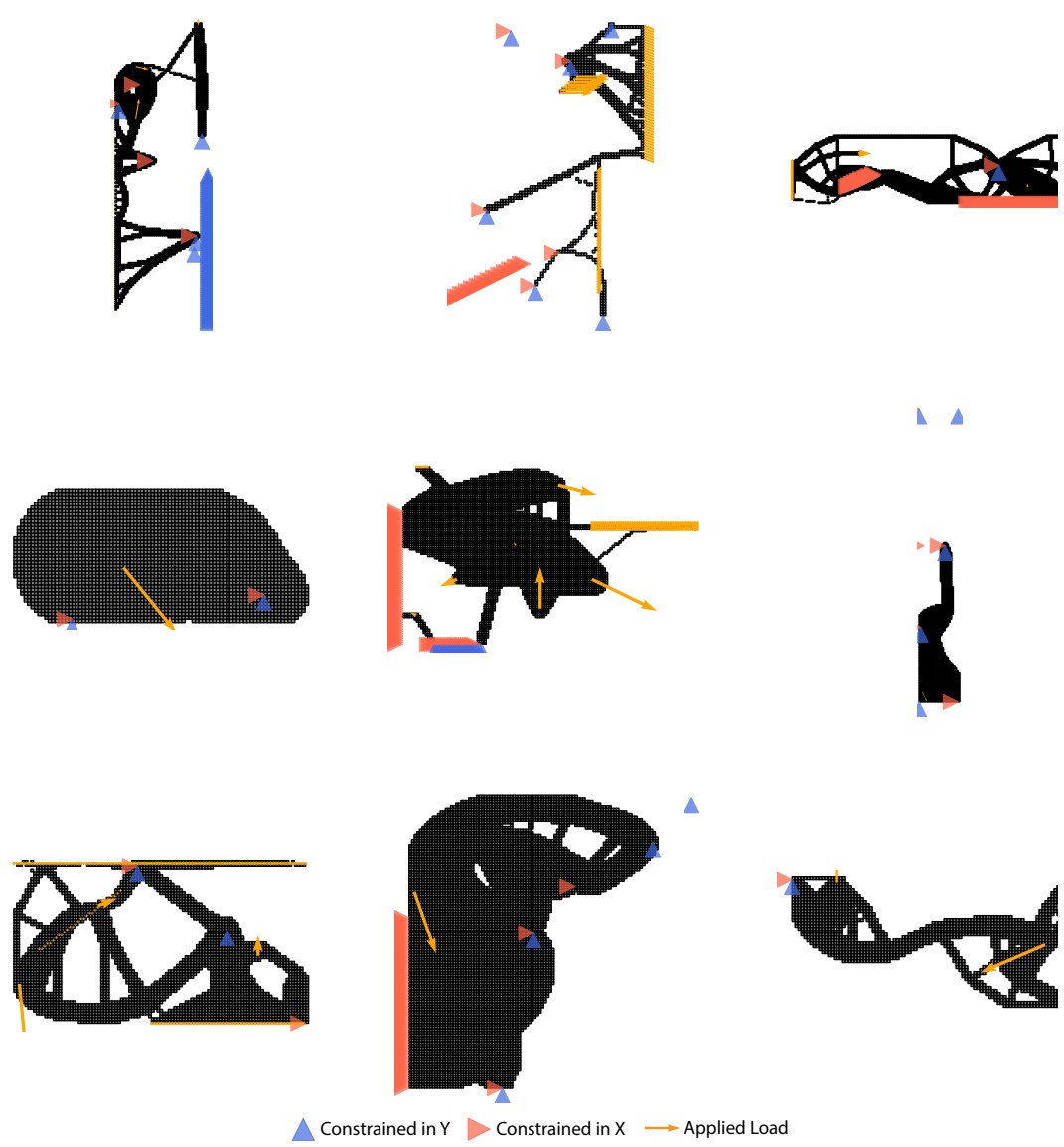

**Figure 23:** Samples from OpenTO Dataset.

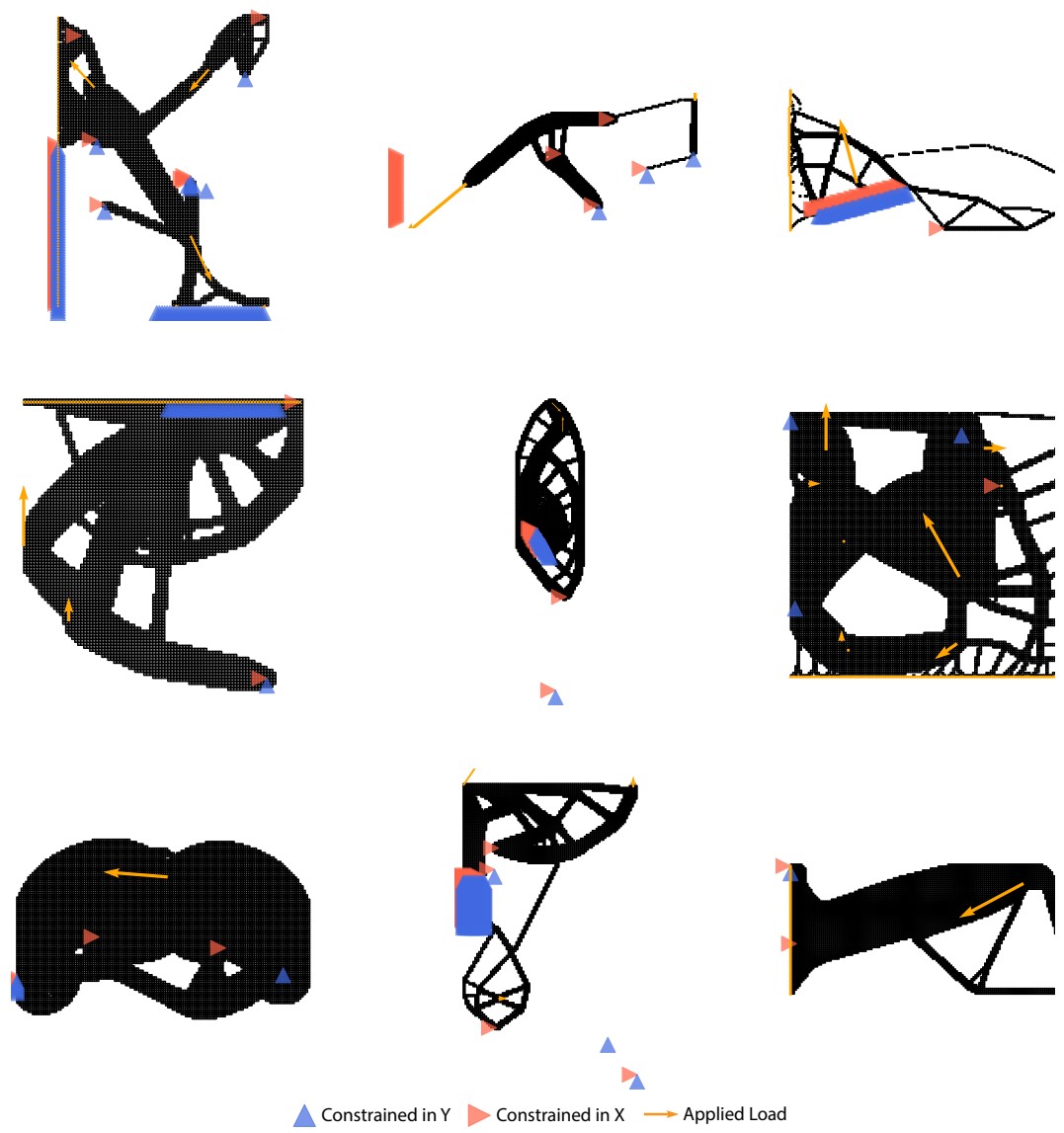

**Figure 24:** Samples from OpenTO Dataset.

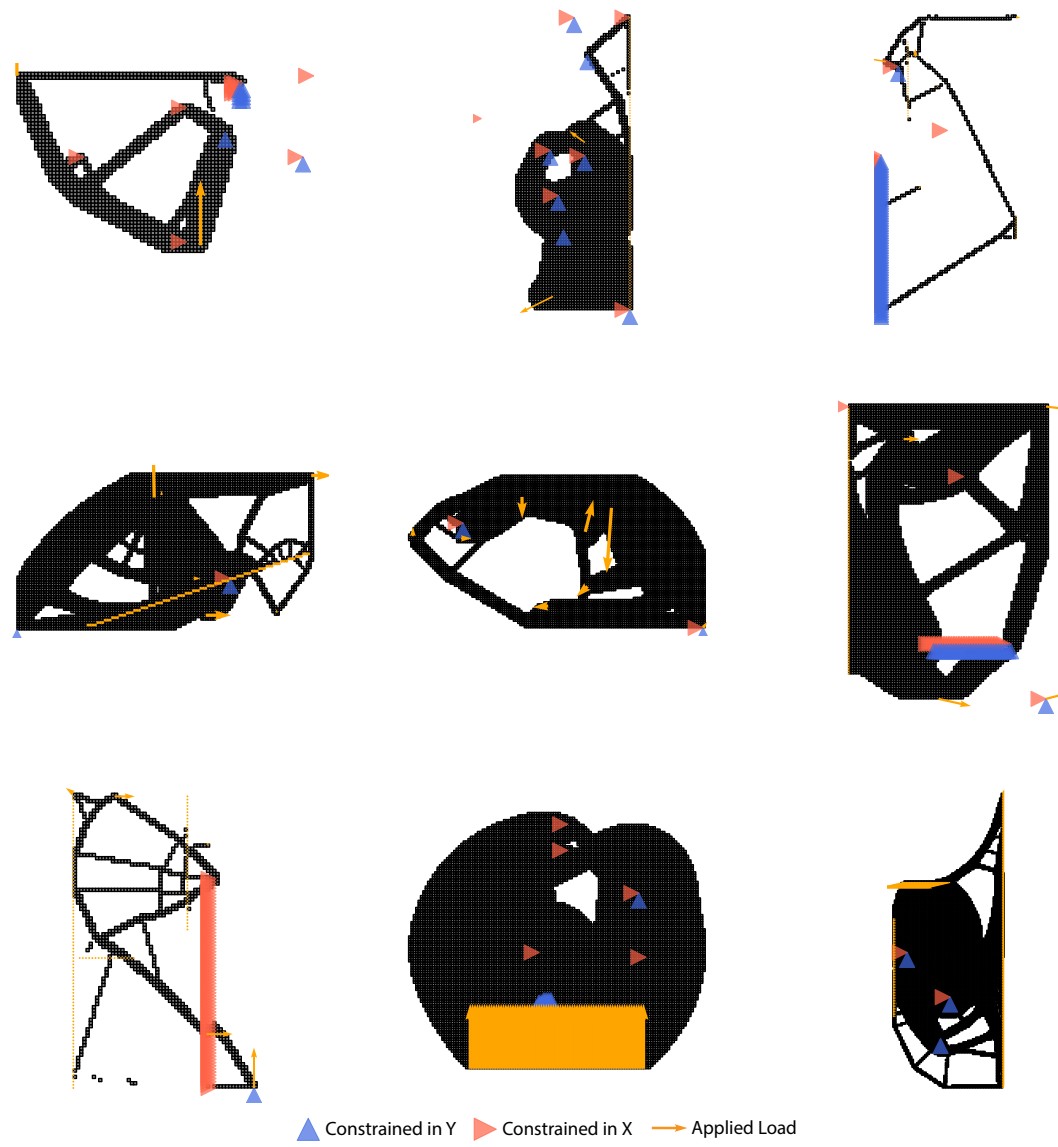

**Figure 25:** Samples from OpenTO Dataset.

### F.7 Combining Prior Data With OpenTO

We generate 2M samples as described above and provide labels (configuration information) for 700,000 samples, and provide 10,000 test samples based on the same procedural generation process. Finally, it is important to note that we also merge the 194,000 data points used in prior work [34] into our dataset, expanding it to 2.2M total samples.

## G Break-Even Analysis

Critics cite the break-even point [62], $\tau = \frac{C_{\text{train}}}{C_{\text{SIMP}} - C_{\text{inference}}}$ (where $C$ is computational cost), which for OAT-like models is dominated by data generation. Our data generation cost (168 H100-days) vastly exceeded our final training run (16 H100-days). Ignoring minor preliminary development runs, the data-generation-dominated calculation yields a break-even of $\tau \approx 2.32$ million uses. This cost seems hard to justify; however, we argue that this metric does not capture the full picture of deep learning's computational efficacy. This is because it ignores the hardware efficiency of parallel data generation and, more importantly, fails to capture the value of democratizing fast TO for users without large compute, enabling greater design exploration and more experiments. This metric points to over 2 million uses to justify a total computational run time of less than one month. Although highlighting high development costs, we believe this argument justifies the large break-even point.

## H Code & Data

The code and data for OAT can be found at: https://github.com/ahnobari/OptimizeAnyTopology.

