# OpenReview forum: "Optimize Any Topology: A Foundation Model for Shape- and Resolution-Free Structural Topology Optimization"
_NeurIPS.cc/2025/Conference — NeurIPS 2025 poster_

### Official Review · Reviewer_zRq4 · 2025-06-23

**Clarity:** 3
**Significance:** 2
**Originality:** 2
**Rating:** 2
**Confidence:** 4

**Summary:**

This paper introduced Optimiza Any Topology (OAT), a foundation model for structural topology optimization (TO) with linear elasticity, together with OpenTO, a new, large-scale dataset with $2.2 \times 10^6$ optimized solutions
across $2 \times 10^6$ unique boundary conditions (BCs). OAT is shown to outperform previous deep learning methods for TO on multiple datasets and is the first generative model to be able to generate solutions for a wide range
of resolutions, aspect ratios, loads, and fixture configurations.

OAT is a latent diffusion model with a neural field decoder that can handle arbitrary resolution outputs, a significant limitation of previous methods like Topodiff. A conditional diffusion model generates variable geometries in
latent space, the problem specifications are encoded using MLPs and Boundary Point Order-invariant MLPs (BPOMs) for point-cloud representations of forces and fixtures.

On multiple benchmarks, OAT demonstrates state of the art (SoTA) performance, reducing the compliance error compared to the numerical baseline method (SIMP) by up to $90\%$ compared with previous models and maintains fast inference times on a single GPU.

**Questions:**

- You don't just consider Dirichlet BC but also Neumann, right? At least in appendix A this should be mentioned at some point for completness.
- Table 5 reports SIMP runtimes: Are these run on the CPU (single core or multiple cores?) or on the GPU?
- Can you provide the exact computational cost and break-even points?
- Failure cases: You report failure cases of OAT (see W2). Do these only appear in OOD test cases? Does OAT consistently fail on these cases or does it produce valid shapes when queried multiple times? Are there (OOD) where OAT fails completely even after multiple queries?

I'm open to discussion and changing my rating if:

- the computational costs/break-evens are included correctly
- the authors address my points from the Weaknesses section.

Open question: OAT is a neural surrogate that approximates the entire TO process in a single step from BC to optimized structure. The expensive part in solving many large TO problems are typically not the SIMP updates but the FEM solves. Where do you see the advantages/tradeoffs between the OAT approach and instead developing a FEM neural surrogate to speed up SIMP?
For me this seems similar to GANs vs Diffusion/Flow matching, where it has turned out to be much better to transform the noise-to-data distribution in multiple steps instead of in a single stepas in GANs.

**Ethical Concerns:**

["NO or VERY MINOR ethics concerns only"]

**Final Justification:**

In my opinion, despite the massive scaling of model and especially dataset size, OAT is still plagued by the same issues as other data-driven TO: a 52% success rate on novel problems (aka 48% failure rate). I feel this core problem is not adequately addressed. I stand by  my concern of overclamining.

I guess I'm just opposed to the approach, unlike the other reviewers.

**Limitations:**

yes

**Quality:**

2

**Strengths And Weaknesses:**

## Strengths
- OpenTO is a new TO training dataset with a diverse set of loading conditions, applied forces, and prescribed volumes.
- OAT is, because it is trained on OpenTO, able to handle a diverse range of BCs/forces.
- A comparison against both SoTA data-driven models (Topodiff, ...) and the standard numerical method (SIMP).
- Tests of OOD problems and documentation of failure cases.
- Paper is well written: Clear structure, good overview, Figure, good Tables

## Weaknesess
- My fundamental criticism: While the results of OAT compared with earlier models like TopoDiff are definitely impressive and a clear step forward, I believe the term "foundation model" is overclaiming. OAT only applies to linear elasticity problems and only to 2D problems; this would all be considered fairly toy-problems by modern SOTA TO research. Compared to foundation models in other science applications (e.g. weather modelling), OAT can't be considered to be a foundation model. And even in this constrained setting it struggles to generalize to OOD cases, as shown in Table 4 (48.55% FR without PP).
 - The failure case (Figure 3, right) looks like OAT has simply memorized it's training dataset and is generating a training datapoint which had loading points close to the test case. Seems likely given the size of OpenTO.
- Lack of proper reporting of computational costs. [1] suggests that DL methods for TO should provide a clear overview of the entire computational costs of generating the training data, training the model and inference: Only inference costs are measured exactly and a rough time given for training (2 days for the autoencoder, 4 days for the diffusion model without accounting for multiple training runs used for hyperparameter tuning etc.).
- [1] further suggests DL TO models should provide breakeven estimates, so how many model queries are needed to amortize the dataset generation and training costs. These breakeven calculations are not provided.

[1] WOLDSETH, Rebekka V., et al. On the use of artificial neural networks in topology optimisation. Structural and Multidisciplinary Optimization, 2022, 65. Jg., Nr. 10, S. 294.

---

> ### Author Rebuttal · Authors · 2025-07-31
>
> We thank Reviewer zRq4 for their thorough review. We value the detailed feedback about defining the foundation model, analysis of failure cases, and the need for computational cost analysis. In the limited rebuttal length, we hope to address each point. (Note, we will refer to the reviewer's reference [1] as [58])
> ## Clarifying The Foundation Model Claims
> We believe we can justify the foundation claims and adjust the paper to better scope and reflect this. According to the widely accepted Stanford definition by Bommasani et al.[1], A foundation model is trained on broad data and can be adapted to diverse downstream tasks [1]. Specifically, the authors mention, "A foundation model is itself incomplete but serves as the common basis from which many task-specific models are built via adaptation". OAT meets the above criteria within the domain of structural topology optimization (STO) and provides diverse paths for downstream adaptations. Below, we explicitly describe how OAT can be deemed a foundation model in this context:
>
> 1. Broad Data: OAT is trained on OpenTO with 2.2M samples covering unique boundary conditions, diverse aspect ratios (1:1 to 10:1), and resolutions (64 to 1024 px). Previous models were trained on an order of magnitude less data.
> 2. Generalization: OAT generates optimized topologies for arbitrary boundary conditions and resolutions without retraining. In contrast, no previous data-driven method has achieved this generalizability (33-34 generalizability score based on [58]).
> 3. Scale and Efficiency: It has 730M parameters and achieves fast inference (<1 second on a single GPU). It also provides >10x reduction in compliance error compared to prior neural methods without post-optimization.
> 4. Adaptability: Like many other foundation models in computer science, OAT provides many paths for downstream adaptation. Like CLIP, a widely regarded foundation model which was trained on large image-text pairs and adapted to various downstream vision-language tasks, OAT’s learned problem embeddings could be used for other applications involving linear elasticity, such as rapid compliance prediction without full TO or other physics-based downstream tasks. This is enabled by the generality of OAT in handling boundary conditions and its large-scale training. Moreover, OAT’s latent diffusion architecture can be adapted. While trained for minimum compliance, it could be fine-tuned or guided for other objectives like stress minimization, making the pre-trained model a powerful starting point. Finally, OAT's autoencoder, trained on 2.2 million topologies, provides a rich latent space for complex topologies. This expensive-to-train component can be reused as a neural operator, like a representation of topologies in classical optimization, or to train smaller, specialized diffusion models for new TO problems at a fraction of the cost.
>
> On the "2D linear elasticity" point, we acknowledge and plan to explicitly clarify that our current foundational claim is scoped specifically to this. Importantly, the Stanford definition does not mandate immediate multi-physics or 3D coverage to be deemed foundational; rather, it emphasizes breadth and adaptability within the targeted domain. The above arguments validate this. Moreover, foundational engineering models often begin with targeted domains (e.g., AlphaFold for protein folding or Finite-Flow FM for CFD) before extending to broader contexts. Indeed, future extensions to 3D or multi-physics represent straightforward adaptations leveraging the existing OAT backbone. We plan to explicitly clarify our claim to " 2D linear elasticity" and will define "foundation model" according to [1] in Section 1 to avoid confusion and scope our claim in Section 6. We hope this addresses the reviewer's main concern, and the reviewer will consider raising their score in light of this proposed clarification.
> ## On Computational Cost and Breakeven
> We appreciate the reviewer for pointing out this criticism of DL methods by the TO community. We do have some concerns about the metric of breakeven, but we see value in adding this analysis to the paper. We propose adding a subsection 5.4 after the inference discussions on breakeven analysis. The proposed contents of this section will be as follows:
>
> >Critics of data-driven TO often bring up the breakeven point [58], $\tau=\frac{C_{\text{train}}}{C_{\text{SIMP}}-C_{\text{inference}}}$ where $C$ is **computational cost**, which for models like OAT is dominated by data generation. Our data generation cost (3 weeks on 8 H100s) vastly exceeds our final training run of 16 H100-days (Appendix E). We ignore the cost of our preliminary development runs—two runs totaling roughly 1/10th of the final training cost (run for 10 epochs at 1/2 resolution)—as the calculation is dominated by data generation, leading to a conventional break-even of $\tau\approx 2.32$ million uses. Note that this metric does not truly capture the full picture of the break-even computational efficacy of deep learning methods. This is because it ignores the hardware efficiency of parallel data generation versus single runs and, more importantly, fails to capture the value of democratizing fast TO for users without large compute, enabling far greater design exploration and more experiments. Given a total computational run time of less than one month, OAT enables fast TO for anyone using our trained model checkpoint, unlocking foundation model capabilities and fine-tuning opportunities. We emphasize that, as with LLMs such as ChatGPT, the one-time expense of producing 2.2 M optimized exemplars is amortized over all future queries. No single user is expected to ‘pay back’ the training cost; instead, the community collectively benefits from near-zero marginal cost per design query once the model is trained.
>
> We hope that this analysis clarifies the scope and contributions of the work. We hope that by clearly showing the break-even point, our justification for it, and clarifying experimental runs and their cost, the reviewer will consider raising their score.
> ## On OOD, Overfitting Concerns, and Sampling
> We thank the reviewer for their valuable feedback. We address the concerns about overfitting and out-of-distribution (OOD) performance by arguing that failures stem from problem sensitivity rather than memorization. We support this with a new experiment.
>
> First, memorization is statistically infeasible.
> 1. **No Data Overlap:** We have verified that no test samples exist in the training set.
> 2. **Combinatorial Space:** The problem space is immense. Even for the smaller problems (4225 nodes in 64x64 configuration), there are $\approx 2^{4225}$ possible boundary configurations, making it highly improbable for our model to find near-matches within only 2M training examples (this does not even consider the fully random domain shape and resolutions, and loads). The visual diversity in Appendix E.6 further supports this.
>
> We hypothesize that OAT generates diverse, nearly-valid solutions, but the high sensitivity of the problem can cause some to fail (Figure 3, which is a test OOD example). To demonstrate this, we performed a new "best of $N$" sampling experiment, generating $N$ solutions per problem and selecting the best.
>
> **Best of N Sampling Experiment (Note: one single run - so numbers are slightly different from Table 4)**
> | N | 1 | 2 | 4 | 8 | 16 | 32 | 64 | 128 | 256 |
> | :--- | :--- | :--- | :--- | :--- | :--- | :--- | :--- | :--- | :--- |
> | Failure Rate | 49.03% | 45.83% | 41.66% | 34.37% | 28.12% | 20.83% | 19.79% | 15.34% | 9.86% |
> | CE* % Mean | 10.85 | 9.51 | 8.71 | 7.33 | 7.59 | 6.34 | 4.39 | 3.19 | 3.08 |
> | VFE* % Mean | 0.92 | 0.99 | 1.04 | 1.11 | 1.18 | 1.25 | 1.33 | 1.37 | 1.35 |
>
> The results show that as we sample more, the failure rate drops from 49% to under 10%, and CE improves. This demonstrates two key points:
> 1.  OAT learns a distribution of plausible solutions, proving it is not memorizing training data.
> 2.  This sampling method is a powerful way to use OAT to find valid, high-quality designs in parallel (a potential alternative to post-optimization).
>
> We will add this experiment to the final manuscript and include a discussion. We believe it resolves the concerns regarding OOD generalization and overfitting.
> ## Response to Minor Points
> We thank the reviewer for their suggestions, which will improve the paper's clarity. To address the missing explicit reference to Neumann boundary conditions and body forces, we will make two updates:
> 1.  In Appendix B, Section B.2 (after line 699), we will add the following sentence:
>     > OpenTO also includes loads which we refer to as distributed loads, which are typically characterized formally as Neumann boundary conditions when loads are a result of stress/pressure applied at the boundaries, and body forces when an internal load is applied to part of the domain interior (often seen in electromagnetic forces or gravitational body forces).
> 2.  In Appendix A (line 636), we will change the wording to "...surface tractions (i.e., Neumann Boundary Conditions)...".
>
> To clarify the SIMP inference times are CPU run times using MKL compiled CHOLMOD as the solver, we will add the GPU times (our own fast GPU multi-grid solver) and update the table caption (this was originally omitted as GPU acceleration in 2D is negligible).
> * The added SIMP-GPU speeds are **9.13s for 64x64** and **68.30s for 256x256**.
> * The following text will be added to the caption:
>     > "SIMP optimizer times are reported using an Intel 14-900K CPU (RTX4090 is used for SIMP-GPU), and all SIMP inference speeds are measured for 150 steps of optimization (which we found was the overall average iteration count to convergence)"
>
> We hope these changes address the raised concerns.
>
> ## Open Question
> Given space, we will discuss this during the discussion period.
>
> [1] Bommasani, R., et al (2022). On the Opportunities and Risks of Foundation Models.

---

> > ### Comment · Reviewer_zRq4 · 2025-08-04
> >
> > - Foundation model: I appreciate the added clarifications. The point remains that you claim in the titel to have a FM for any shape/topology optimization, but do in fact only have it for 2d linear elasticity. It is not clear to me and not shown that it can be easily fine tuned to e.g. TO for flow problems [2] or photonics [3] which is governed by very different equations.
> >
> > - Generalization: OAT still has a 48% failure rate without preprocessing, to me this indicates some issues in generalization ability remain. The added experiment of generating multiple shapes and the large reduction in failure cases is very valuable and mitigates my concerns to some extent.
> >
> > - Computational cost/Breakeven: Thanks for this addition, I think add this discussion is important as this point has been raised by TO researchers.
> >
> >
> > [2] Gersborg-Hansen, Allan, Ole Sigmund, and Robert B. Haber. "Topology optimization of channel flow problems." Structural and multidisciplinary optimization 30.3 (2005): 181-192.
> > [3] Jensen, Jakob Søndergaard, and Ole Sigmund. "Topology optimization for nano‐photonics." Laser & Photonics Reviews 5.2 (2011): 308-321.

---

> ### Author Response · Authors · 2025-08-04
> **Further Discussion Part 1**
>
> Thanks for your timely response and giving us the opportunity to respond to your comments. We would like to make some concrete points regarding both the foundation model matter and generalizability concerns brought forth by the reviewer. We hope we are convincing and clear on both terms, and will convince the reviewer.
>
> ## Further Discussion On Foundation Model Clarification
> We again like to start by highlighting the well-understood notion of foundation models. As discussed in our rebuttal, the common definition and analysis of foundation models clearly concedes, "A foundation model is itself incomplete but serves as the common basis from which many task-specific models are built via adaptation"[1], and as we mentioned in the rebuttal, we have no contention on the fact that OAT is foundational in the scope of Linear Elasticity as the primary physics and as mentioned, despite having made this clear in the paper are very much happy to very explicitly add "2D linear elasticity" as the scope in both section 1, abstract and conclusion (Note that this is already been clarified in the paper on lines 71( ... Minmum Compliance ...), 75-76 (Scopes to arbitrary shape resolution explicitly), and 414-416 (Explicitly mention that future work is needed to expand to other physics and multi-physics) and we just like to remove any confusion on claims and limitaitons). Aside from this, we would like to further clarify how the adaptability of OAT is valuable as a foundational framework for adaptation to other tasks, both within Linear Elasticity and **Beyond**.
>
> **Out-of-the-Box Auto-Encoder Adaptability to Other Physics**
> As the paper title suggests, we are the first model to introduce a generative framework for resolution and shape-adaptable TO. This is enabled by the auto-encoder (AE) and the generalizable latent representation that the large-scale training on the 2.2M OpenTO dataset enables. The pre-trained latent space model is built for resolution- and shape-free reconstruction without any explicit physics, thus we blieve and as we show below it is not necessarily limited to linear elasticity. The auto-encoder is applicable to arbitrary shape and resolution 2D topologies with binary density fields, and we have shown that the reconstruction is capable of handling many different topologies (Appendix E.6 shows the diversity of OpenTO samples).
> While our initial claim is clearly around shape- and resolution-free TO in minimum compliance and linear elasticity, inspired by the reviewer's comment, we conducted a quick physics generalizability test below, where we show the latent space model does indeed generalize to different types of TO. We were able to obtain the dataset used in [2] for heat-sink topology optimization using their public repository, which involves different physics (Heat Conduction) and a largely arbitrary[2] 71x71 mesh, which our model is not explicitly trained on (both the physics and resolution). Without re-training or fine-tuning, nor the need for resizing these topologies (an important feature for generalization in OAT), we ran the AE in our model to reconstruct these vastly different-looking topologies (see Figures 1,4, and 5 in [2] for visualizations) and measure an **average IOU of 0.9408** with a SD of 0.04. This shows that out-of-the-box (zero-shot, even without fine-tuning or retraining), the latent representations of OAT are a powerful foundation for training diffusion models for different physics and geometries without the need for expensive, large-scale data. While 0.94 IoU is very good, one could potentially improve this further by fine-tuning OAT quickly on small data to even better reconstruct topologies in different physics.
> We did not add such experiments to the paper as the initial claim was built around shape- and resolution-free generalization in linear elasticity, and we see multi-physics generalization as future work. If the reviewer requests, we are happy to add this experiment to the appendix, although we see this as out of scope, as we did not claim multi-physics generalization in the paper.
>
> **In Scope Adaptability:**
> A summary of the original rebuttal is that OAT is adaptable in three ways. 1. OAT's boundary condition embeddings can be repurposed for other physics-based tasks within linear elasticity (e.g., predicting what optimal compliance will be for a given problem), and 2. its latent diffusion architecture can be fine-tuned for new objectives like stress minimization and 3. the first of its type shape- and resolution-free autoencoder component discussed above can be reused independently as a neural operator or to train smaller, specialized models at a fraction of the cost.

---

> ### Author Response · Authors · 2025-08-04
> **Further Discussion Part 2**
>
> ## Clarifying Precision Performance
> First, we want to highlight that OAT has a 100% success rate (0% failure) and a major performance boost (90% reduction in compliance error compared to SOTA) on the existing benchmarks, which have been popular in prior works [3,4,5,6]. However, we introduce the OpenTO benchmark, which goes beyond these limited benchmarks, and thus, the point raised can be addressed by two discussion points.
>
> 1. **Benchmark Contribution:** The TO community has often criticised DL approaches for TO by bringing up the generalizability issue with respect to resolution and shape, as well as arbitrary boundary conditions, both of which are missing in any prior benchmarks commonly used. OpenTO provides a benchmark that is fully general (33-35 out of 35 generality score based on [58]), which makes it a true challenge that the TO community has been asking for and OAT is built to address this matter; thus, our work addresses this generalizability issue, both from a benchmark and method perspective. This makes the OpenTO benchmark leaps ahead of other benchmarks, and thus represents a new paradigm that has to be taken into account when judging performance. These tough benchmarks pave the way for greater and more general foundation models, and thus performance in such benchmarks is not expected to be at or near 100% (e.g., even a model as large as GPT-3.5 performed at 48% on HumanEval code benchmark[7])
> 2. **OAT Performance on Benchmarks:** As mentioned before, OAT significantly outperforms on existing benchmarks and does not exhibit failure in these popular albeit simpler benchmarks. Furthermore, despite the challenging nature of the OpenTO benchmark, OAT has a 52% 1-shot success rate, which with more sampling (that comes at a rather low time cost in generative models) achieves much greater performance, which demonstrates that the samples OAT generates are near valid, and its the high problem sensativity to very small details that lead to such failures (See Figure 3), thus we believe that OAT performs rather well considering inference-time scaling (which we thank the reviewer for acknowledging) and although can be improved in future works, presents the first generative model to even approach such a challenging problem, thus making it an important contribution to the community and presents a viable answer to the generalization criticisms of the TO community.
>
>
> ---
> We hope that the response to the remaining two concerns helps the reviewer see the impact of the work for the community and increase their score.
>
> ---
>
> [2] Bernard, S, Wang, J, & Fuge, M. "Mean Squared Error May Lead You Astray When Optimizing Your Inverse Design Methods." Proceedings of the ASME 2022 IDETC-CIE. St. Louis, Missouri, USA. August 14–17, 2022. V03AT03A004. ASME. https://doi.org/10.1115/DETC2022-90065
>
> [3]Kuszczak, I., Kus, G., Bosi, F., & Bessa, M. A. (2025). Meta-neural Topology Optimization: Knowledge Infusion with Meta-learning.
>
> [4]Nie, Z., Lin, T., Jiang, H., & Kara, L. B. (2020). TopologyGAN: Topology Optimization Using Generative Adversarial Networks Based on Physical Fields Over the Initial Domain.
>
> [5]Giannone, G., Srivastava, A., Winther, O., & Ahmed, F. (2023). Aligning Optimization Trajectories with Diffusion Models for Constrained Design Generation. Thirty-Seventh Conference on Neural Information Processing Systems.
>
> [6]Dhariwal, P., & Nichol, A. Q. (2021). Diffusion Models Beat GANs on Image Synthesis. In A. Beygelzimer, Y. Dauphin, P. Liang, & J. W. Vaughan (Eds.), Advances in Neural Information Processing Systems.
>
> [7]OpenAI, Achiam, J., et. al. (2024). GPT-4 Technical Report.

---

### Official Review · Reviewer_9u8P · 2025-06-23

**Clarity:** 4
**Significance:** 3
**Originality:** 3
**Rating:** 5
**Confidence:** 3

**Summary:**

Topology optimization is the task of optimizing an engineering design with an evaluation metric that depends on the solution of a PDE, such as the optimization of an airfoil to minimize the drag resulting from the flow field obtained by solving the Navier-Stokes Equations. Such optimization is typically costly and is performed with expensive adjoint-based optimization. In place of this, the paper trains a foundation model for topology optimization for a variety of domain sizes and shapes. In the process, they also develop an extensive dataset of topology optimization tasks and corresponding solutions.

**Questions:**

- It seems like data were generated at random in the proposed OpenTO dataset through procedural generation; however, it was not described how this generation is performed. In particular, I would imagine many domain with purely random generation would be implausible, so what are the mechanisms to cull or avoid these domains?

**Ethical Concerns:**

["NO or VERY MINOR ethics concerns only"]

**Final Justification:**

My initial read of the paper suggested that both the proposed foundation model and dataset were worthwhile contributions to the broader community. Given that there are not many *conceptually* novel aspects of the work (in the sense that each part of the OAT foundation model has been leveraged in other models in the past and similarly for the data generation process), I was initially uncertain of how worthwhile the contribution to the broader community would be. However, reading the reviews from reviewers more intimiately familiar with the empirical side of this community, it seems like these contributions are wholly worthwhile despite the conceptually straightforward nature of it.

I am, therefore, increasing my score to reflect this.

**Limitations:**

Yes

**Quality:**

3

**Strengths And Weaknesses:**

The primary contribution from this paper is a generally usable dataset of topology optimization tasks. While the component parts of the proposed model are standard, the assembly into a foundation model likely required substantial effort to get functioning, as we elaborate on below.

**Quality**

The bulk of the work was in considering extensive empirical assessments of the proposed model against other topology optimization approaches. The authors do highlight well the deficiencies in their proposed model, which is more broadly an issue of using black-box machine learning predictors for directly predicting the solutions to such optimization tasks. The assessments are thoroughly conducted against number of different generated benchmark tasks, assessing for the compliance, failure rate, and sampling speed. These three metrics seem quite comprehensive as metrics for assessment and were assessed against some competitor methods.

**Clarity**

The presentation is well-delivered and writing clear. Figures are also well-illustrated, such as the workflow diagram in Figure 2 and failure diagram in Figure 3. Exposition was easy to follow along throughout the paper.

**Significance**

The work does contribute significantly in developing a comprehensive suite of topology optimization problems. This alone seems to be a worthwhile contribution, although the additional contribution of the foundational model serves as a good starting point for further exploration in this direction. The pointing out of deficiencies in this direct prediction approach around boundary effects is also a worthwhile contribution.

**Originality**

While no individual piece of the pipeline is inherently novel or distinct from their incarnations in previous works, the overall construction of a topology optimization foundation model from these pieces was a novel undertaking as was the generation of plausible topology optimization problems for training the model. More novelty in the foundation model would be interesting to pursue, but this feels like work that would be better suited to further work that builds upon this first step.

---

> ### Author Rebuttal · Authors · 2025-07-30
>
> We thank Reviewer 9u8P for their balanced review and for recognizing the significant effort involved in creating both the dataset and the model. We appreciate the positive comments on the paper's clarity and the thoroughness of our empirical assessments. We will provide a more detailed explanation of the procedural generation and curation process for the OpenTO dataset to address the excellent question raised, and we hope that the rebuttal and the new experiment provided in response to other reviewers will convince the reviewer to raise their score.
>
> ## On Procedural Data Generation
> We thank the reviewer for bringing up this important point of discussion. We agree that it is important to be very clear on how the data was generated and what pipeline was used for OpenTO. We omitted the full discussion on the generation process in the main body of the paper because this involves many different nuances and details, which would not benefit the readers immediately, and we believe this would have lengthened the paper and reduced its readability. As such, in section 4.5 on the dataset, we briefly describe the features of the dataset and refer the readers to Appendix B, where the full details of the procedural generation are given. In Appendix B, we describe in full detail how OpenTO was created. On your concern about many domains being physically invalid, this is very much true in many random configurations, which is why in section B.3 we describe the validity check for a randomly generated problem. Mainly, we look at whether the generated configuration is physically fully constrained and the resulting FEA matrix is non-singular, thus making the problem solvable.  Beyond this physical feasibility check, we do not perform any other curations and make this deliberate choice informed by other works in deep learning for engineering design tasks, which show that fully random data, so long as physically valid, provides a great pathway for developing models that capture the fundamental physics and engineering requirements. A few examples of such datasets that have been procedurally generated are datasets for meta-materials[1], linkage mechanisms[3], or, more recently, the Alpha geometry dataset, procedurally generated by Google DeepMind [3] for geometry problems. The success of these validated yet not deeply curated datasets informed our decision to limit curation to only physically valid configurations, rather than curating for realism or relatedness to real-world problems. This is simply because we hope OAT can serve as a pre-trained foundational model that can be adapted to many downstream tasks, and thus, capturing the underlying physics is of great importance, and we believe that less curation could be helpful in this kind of generalization. We hope that this and the existence of the detailed procedural generation process in the appendix have cleared the concerns of the reviewer, and the reviewer would consider raising their score if they are satisfied with our response to this concern.
>
> [1] Lee, D., Chan, Y., Chen, W. (., Wang, L., van Beek, A., and Chen, W. (November 3, 2022). "t-METASET: Task-Aware Acquisition of Metamaterial Datasets Through Diversity-Based Active Learning." ASME. J. Mech. Des. March 2023; 145(3): 031704. https://doi.org/10.1115/1.4055925
> [2] Heyrani Nobari, A, Srivastava, A, Gutfreund, D, & Ahmed, F. "LINKS: A Dataset of a Hundred Million Planar Linkage Mechanisms for Data-Driven Kinematic Design." Proceedings of the ASME 2022 International Design Engineering Technical Conferences and Computers and Information in Engineering Conference. Volume 3A: 48th Design Automation Conference (DAC). St. Louis, Missouri, USA. August 14–17, 2022. V03AT03A013. ASME. https://doi.org/10.1115/DETC2022-89798
> [3] Chervonyi, Y., Trinh, T. H., Olšák, M., Yang, X., Nguyen, H., Menegali, M., … Luong, T. (2025). Gold-medalist Performance in Solving Olympiad Geometry with AlphaGeometry2. arXiv [Cs.AI]. Retrieved from http://arxiv.org/abs/2502.03544

---

> ### Comment · Reviewer_9u8P · 2025-08-03
>
> I thank the authors for their thoughtful response. After reading through the response and points raised by the other reviewers, I am willing to raise my score, seeing this paper as a worthwhile contribution to the community.

---

> > ### Author Response · Authors · 2025-08-04
> >
> > Thank you very much for devoting your time to reading through the reviews and our response. We are pleased to see that you have found our rebuttals thoughtful and are willing to raise your score.

---

### Official Review · Reviewer_77iF · 2025-07-02

**Clarity:** 3
**Significance:** 3
**Originality:** 3
**Rating:** 5
**Confidence:** 4

**Summary:**

The paper proposes OAT (Optimize Any Topology), a resolution-invariant framework that fuses a shared autoencoder, a convolutional local implicit-field renderer, and a conditional latent diffusion model to synthesize near-optimal topology designs for arbitrary aspect ratios, resolutions, volume fractions, loads, and constraints. Leveraging the 2.19 million-example OpenTO dataset covering 2 million distinct boundary conditions, OAT attains sub-1 % compliance error on 64x64 and 256x256 datasets and cuts inference time to faster than SIMP and roughly more accurate than prior neural baselines such as NITO and TopoDiff.

**Questions:**

see weaknesses.

**Ethical Concerns:**

["NO or VERY MINOR ethics concerns only"]

**Final Justification:**

I believe this paper presents a valuable new dataset and proposes a correspondingly more general method, so I have a positive view of the work. The author's rebuttal has also largely addressed the concerns I raised. However, since I am not very familiar with this field, it is difficult for me to justify increasing my score further.

**Limitations:**

see weaknesses.

**Quality:**

3

**Strengths And Weaknesses:**

**Strengths**

1. **Large, publicly released dataset.** The paper introduces the OpenTO corpus, containing 2.19 M samples and 2 M distinct boundary conditions, which offers the community an unprecedentedly broad and reproducible benchmark for training and evaluating topology-optimization models.

2. **State-of-the-art accuracy.** On $64\times 64$–$256\times 256$ datasets, OAT achieves compliance errors below $1\%$ without post-optimization, outperforming prior generative baselines such as TopoDiff and NITO by roughly one order of magnitude.

3. **High versatility.** Thanks to its resolution-invariant latent representation, OAT handles arbitrary aspect ratios, mesh sizes, volume fractions, loads, and boundary constraints without retraining, markedly simplifying deployment across heterogeneous design scenarios.


**Weaknesses**

1. **Attribution of gains to the new dataset is unclear.** OAT is evaluated on OpenTO, whereas competing methods use their original training data; the reported superiority may therefore stem from higher-quality data rather than algorithmic advances. A fair comparison would retrain existing generative approaches on OpenTO (minor architectural adjustments should suffice) to isolate the methodological benefit of OAT itself.

2. **High failure rate under fully random conditions.** A single forward pass yields $\approx48$% invalid topologies on the unrestricted OpenTO benchmark, and even after ten SIMP refinement steps the failure rate remains $\approx16$%, which may limit reliability in safety-critical workflows.

3. **Dependence on post-optimization.** The recommended 5–10 SIMP iterations needed to “heal” minor deviations contradict the ambition of one-shot optimal generation and diminish real-time performance in high-throughput settings.

4. **Complex architecture and integration cost.** Combining an autoencoder, an implicit-field renderer, and a conditional diffusion model increases engineering overhead and GPU-memory management complexity compared with single-stage UNet- or NeRF-based generators, potentially impeding adoption in industrial pipelines.

---

> ### Author Rebuttal · Authors · 2025-07-30
>
> We thank Reviewer 77iF for their encouraging review and for highlighting how OAT introduces a versatile platform for deep learning in TO problems and achieves SOTA performance by a large margin. We also appreciate the recognition of OpenTO as a valuable dataset. We appreciate the insightful points regarding the performance gains and dataset, failure rates, and model complexity. These are important considerations that we hope to address here to provide a better understanding of our method's capabilities and limitations.
> ## SOTA Of-The-Shelf Models Can't Adapt To OpenTO
> The question of using existing methods with the OpenTO to extract dataset gains compared to architectural gains is a noteworthy one. To respond to the reviewer briefly, adapting existing methods other than NITO is not a simple matter of training them on the new data. This is why we believe OAT is truly the first model to attempt to solve the most generalized variant of the TO problem. And in the case of NITO, we indeed provide experiments of a NITO variant as large as OAT trained on OpenTO, which, as clearly evident in Figure 3 and numerous examples in Appendix D, leads to very low-quality samples, further highlighting the strength of OAT as an architecture. There are primarily two reasons why an adaptation of SOTA is not very simple, and why the need for OAT arises in the first place. Firstly, existing approaches work on fixed resolution and square domains. Secondly, existing methods simply do not permit easy adaptation of the widely varying boundary conditions and forces that OpenTO includes.
>
> ### SOTA Struggles With General Aspect Ratio and Resolution
> One of the major hurdles in most prior works is their inability to generalize to random resolutions. Note that the OpenTO dataset is not just random in its aspect ratios, but it also has varying resolutions, which can reach up to 1024 pixels in a row or column. This means that one must train existing methods such as TopoDiff or DOM at their native $64\times64$ resolution by resizing and centering everything, which, given the very low resolution of $64\times 64$ (minimum pixel count in OpenTO), would most likely yield very poor quality samples and thus not provide a fair comparison. Alternatively, the models would have to be scaled to a larger resolution, such as $256\times 256$, for which training a diffusion model is highly costly and in most cases not even effective at generating high-quality samples, see [31] and their experiments on full diffusion training at a resolution $256\times 256$. This resolution is going beyond most SOTA diffusion models for image generation, e.g., stable diffusion, which tackles this matter by reducing the size of the diffusion space using an AutoEncoder, such as in our approach. Therefore, we truly must emphasize that existing frameworks are using off-the-shelf diffusion models, which, simply put, are not easily adapted to the OpenTO dataset, which involves fully random resolution and aspect ratios. Despite this, we do train a larger version of NITO, which can be adapted to this data and show that the vast diversity of OpenTO hinders its performance in generating high-quality samples, while our proposed framework is capable of producing high-quality, realistic topologies.
>
> ### Boundary Conditions and Problem Representations Are Non-Trivial
> The second major hurdle in adapting existing methods lies in how they represent TO problems. We briefly allude to this issue in the paper (lines 163-169). All prior works, except NITO, represent TO problems in a peculiar manner. Rather than the raw representations of the boundary conditions and loads, and volume fraction, prior works use image-based representations. Specifically to represent boundary conditions, these prior methods run a single FEA simulation on the domain with full density (all pixels black with density 1) and use the stress and strain fields as a representation of boundary conditions in image-like form. This means they compute said fields and pass them as a channel in their diffusion or GAN models. Given that OpenTO has a fully variable shape and resolutions, obtaining consistently sized fields for these different problems will not be trivial, as one has to interpolate and pad such fields to a fixed resolution, which may lose significant fidelity. Furthermore, prior works are designed for simple problems with a single load and finite repeated boundary settings (42 boundary configurations), which makes it easier to decipher the boundary conditions from physics fields. In OpenTO, just looking at stress fields may not provide such granular information for highly complex and non-regular boundary configurations and shapes. Furthermore, one of the reasons prior works have slower inference is their need for one FEA run to represent boundary conditions, which OAT overcomes. [31] discusses this matter in more detail if the reviewer is interested in a more in-depth analysis of this issue.
> ## A Different Viewpoint On High Failure Rates
> We acknowledge that the high failure rate can be a major limitation and agree that it should be addressed. However, we must highlight that we have conducted an inference-time scaling experiment on best of $N$ sampling, which highlights how this matter can be handled. To this end, we have explicitly discussed this as a major pain point for future work (see lines 410-415) and made the case that RL frameworks, which are becoming a common theme in many foundation models, can hopefully aid in improving this aspect of our model. The reason we believe RL frameworks will be effective in this scenario is that we see in examples such as those shown in Figure 3 that the model is capable of producing high-quality samples; however, small inaccuracies are the major reason for the high failure rate. However, our model can be sampled many times for each problem at a very low cost compared to SIMP (Table 5). Thus, we conduct this new best of $N$  experiment, which we plan to add to the manuscript. In this experiment, we sample OAT $N$ times and report metrics on the best of the $N$ for each individual sample in the test set (Not the whole run). Below are the results of this experiment:
>
> **Best of N Sampling Experiment (Note: one single run - so numbers are slightly different from Table 4)**
> | N | 1 | 2 | 4 | 8 | 16 | 32 | 64 | 128 | 256 |
> | :--- | :--- | :--- | :--- | :--- | :--- | :--- | :--- | :--- | :--- |
> | Failure Rate |  49.03 | 45.83 | 41.66 | 34.37 | 28.12 | 20.83 | 19.79 | 15.34 | 9.86 |
> | CE* % Mean |  10.85 | 9.51 | 8.71 | 7.33 | 7.59 | 6.34 | 4.39 | 3.19 | 3.08 |
> | VFE* % Mean |  0.92 | 0.99 | 1.04 | 1.11 | 1.18 | 1.25 | 1.33 | 1.37 | 1.35 |
>
> We see that as we sample OAT more, we reduce the failure rate significantly, verifying the above claim of small inaccuracies causing failures (even a single missed pixel can miss a boundary condition), and showing how RL and rejection sampling refinements of OAT can be effective at reducing the failure rate and improving performance.
>
> We hope the addition of this new experiment addresses the primary concern raised by the reviewer and provides a rich perspective on this limitation of OAT.
> ## Post Optimization Cost Is Much Less Than Full Optimization
> Similar to our discussion on the high failure rate, we do explicitly acknowledge that this additional optimization does come at a cost of slower inference. However, we would like to highlight that in optimization, typically 200-300 steps of optimization are needed for convergence, given the non-linear and non-convex nature of the problem with sensitivity to initialization and the underlying non-linear optimizer. Specifically, looking at Table 5 of the paper, we do note that although post optimization does increase the cost of inference, it still provides an order of magnitude acceleration compared to the conventional SIMP optimizer. However, in our discussion of the efficiency, we do note that this use of post optimization does come at a notable cost, especially at higher resolutions, and agree with the reviewer on this limitation. We do believe that, like the proposed future work on how the failure rate can be addressed, the use of rejection sampling and RL-inspired policy optimization frameworks can be effective at raising OAT's performance on this benchmark. As shown above, although best of $N$ does not significantly reduce compliance error like it does failure, there is a trend towards reduction of error, which gives hope for further improvements in RL, like policy optimization. Noting this is not immediately clear and discussed in our conclusion, we propose a minor change to the language of the conclusion around this future work (aside from adding the new experiment) to put into perspective the potential downstream improvement that OAT provides. Specifically, we propose changing the sentence starting on line 411:
>
> >Firstly, we see that currently OAT faces a notable failure rate and relies on post optimization on fully random benchmarks, which future work will focus on addressing through promising directions such as reinforcement learning[33,43] based on physics models and optimizer guidance in diffusion models[9]
>
> We hope the discussion above satisfies the reviewers' concerns on this matter.
>
> ## On Complex Architecture and Integration Cost
> We agree with the reviewer that OAT introduces a deep learning framework into the workflow. However, we argue that this integration cost is a minor engineering hurdle compared to the significant existing complexity of setting up the FEA, physics, and optimization for any TO problem. Since our provided code and checkpoints can readily use this existing setup, the effort should be minimal. Given the significant acceleration OAT provides (Table 5) and the increasing trend of integrating deep learning into commercial engineering software, we believe the benefits justify this cost. We hope this clarifies the balance of integration cost and inference acceleration.

---

> > ### Comment · Reviewer_77iF · 2025-08-03
> >
> > Thank you for the author's response. I believe it has addressed the issues I raised. Considering the overall contributions of the paper, I will maintain my original score.

---

> > > ### Author Response · Authors · 2025-08-04
> > >
> > > Thank you for your timely response and acknowledging the rebuttal points. We are happy to see that you were satisfied with our responses.

---

### Official Review · Reviewer_ukV1 · 2025-07-03

**Clarity:** 3
**Significance:** 4
**Originality:** 4
**Rating:** 5
**Confidence:** 4

**Summary:**

This paper presents a foundation framework of structural topology optimization by handling arbitrary boundary conditions, domain shapes, and resolutions. It is a pre-trained latent diffusion model designed for minimum-compliance structural design, marking a foundational advance. In this paper OpenTO Dataset is established including 2.2M samples with arbitrary boundary conditions, shapes, resolutions and forces.  It achieves a 10x reduction in compliance error compared to prior methods. Generates high-resolution topologies maintaining efficiency on the inference.

**Questions:**

1. E_min in Sections A.1.3 and A.1.4 should provide a more detailed discussion of their interrelationship.
2. The mathematical symbols used in the vθ formula (line 223) describing the inverse denoising process require clarification. Would you please provide a detailed explanation of this formulation?
3. In the experimental results, the mean volume fraction error (VFE) was reported. However, another metric VFE (Med) (median volume fraction error), was also used in Reference [31] but not mentioned in this study. Could the authors explain why this metric was not included?

**Ethical Concerns:**

["NO or VERY MINOR ethics concerns only"]

**Limitations:**

yes.

**Paper Formatting Concerns:**

The format is good enough.

**Quality:**

4

**Strengths And Weaknesses:**

Strengths:
1. This topology optimization model can handle arbitrary boundary conditions, resolutions, and aspect ratios, overcoming limitations of prior fixed-grid approaches.
2. This model achieves very low compliance error without the need for any direct optimization. It is efficient to generates high-resolution topologies with scalable inference.
3. This paper introduces an OpenTO dataset which contains 2.2 millions structures. The procedural data generation is very clear in Appendix B.
4. This method could handle mixed shape/resolution data easily.

Weaknesses:
 The computational complexity of the method remains uncharacterized from a mathematical perspective.

---

> ### Author Rebuttal · Authors · 2025-07-30
>
> We thank Reviewer ukV1 for their positive assessment and rating. We are grateful for the acknowledgment of our model's ability to generalize to arbitrary conditions with high accuracy and the value of the OpenTO dataset. We will address the specific questions regarding the mathematical details and the VFE to help address the reviewer's concerns and improve the paper. We hope the reviewer will be satisfied with our explanation and proposed modifications.
> ## Calrifying $E_{\text{min}}$
> In TO formulation $E_{\text{min}}$ is added to the overall Young's modulus so that in parts of the mesh or design space where no density is assigned, i.e. $\rho_e=0$ the local stiffness does not become zero, resulting in a singular matrix. To prevent this, a very small $E_{\text{min}}<<E_{\text{solid}}$ is added to prevent absolute zero stiffness in any part of the mesh. In our data generation process and evaluations of all methods, we set $E_{\text{min}}=10^{-9}$, while $E_{\text{solid}} = 1$.
> Since there may be a lack of clarity on this matter, we propose to add a sentence to the final camera-ready version to address this. Specifically, we propose adding the following changes:
>
> In Appendix A.1.3, after line 658, we will add this clarification:
>
> > Note that the choice of $E_{\text{min}}$ is purely to avoid singular values and is selected to be $10^{-9}$ in our data generation compared to the much larger value of $E_{\text{solid}}=1$. This choice is informed by prior literature recommending such a value as a good balance between stable optimization and accuracy of simulations [47].
>
> Further, we will add in Appendix A.1.4, after line 671, we will add:
> > Since in out implemtation we set $E_{\text{min}}=10^{-9}$, the choice of $\rho_{\text{min}}=0$ is used, although numerically any design element in a mesh which recieves a 0 density will effectively have a Young's modulus of $10^{-9}$ which given our choice $E_{\text{solid}}=1$ would be equivalent to setting $E_{\text{min}}=0$ and $\rho_{\text{min}}=10^{-9}$.
>
> We hope that these additions clarify the exchangeability of setting a lower bound above zero on the densities or setting a zero lower bound and the same value as the minimum Young's modulus when $E_{\text{solid}}=1$. Again, these choices are the common values used in most SIMP implementations, and we hope that clarifying this will help with replicating any data generation efforts.
> ## Detailing The Classifier Free Guidance Formulation
> We thank the reviewer for their feedback. We are happy to add the full detailed explanation of the classifier-free guidance scaling.
> We propose changing the sentence (line 222-224): "We can apply this during inference, with guidance scale $w$, by performing the inverse denoising process with $\hat v_\theta(z_t,t, P)\;=\;v_\theta(z_t,t \mid \varnothing)\;+\;w\;\Bigl(v_\theta(z_t,t \mid P)- v_\theta(z_t,t, \varnothing)\Bigr)$, rather than $v_\theta$, which allows us to adjust the guidance strength." to the better clarify the inference with classifier-free guidance.
>
> The proposed replacement is as follows:
> > Classifier-free guidance can be applied in our approach by adjusting the denoising velocity. Normally, in conditional denoising, one would use the predicted velocity at time step $t$, $v_\theta(z_t,t \mid P)$, which predicts the velocity for the current noisy sample $z_t$ given problem embedding $P$. When applying classifier-free guidance we can also compute the un-conditional denoising, namely $v_\theta(z_t,t \mid \varnothing)$, and rather than just using $v_\theta(z_t,t \mid P)$, we measure the shift in velocity direction as a result of conditioning, $v_\theta(z_t,t \mid P)-v_\theta(z_t,t \mid \varnothing)$ and amplify this shift by an over-relaxation factor, $\omega$, leading to a final denoising velocity $\hat v_\theta(z_t,t, P)\;=\;v_\theta(z_t,t \mid \varnothing)\;+\;w\;\Bigl(v_\theta(z_t,t \mid P)- v_\theta(z_t,t, \varnothing)\Bigr)$, which can be used in place of $v_\theta(z_t,t \mid P)$, mimicking what Ho and Salimans [15] propose for classifier-free guidance.
>
> We thank the reviewer for helping improve the clarity and readability of the paper and hope that the above explanation is satisfactory.
> ## Inference Time Provides A Better Measure Of Efficiency
> We thank the reviewer for pointing out formal computational analysis in our work. We believe the reported inference times are a much more practical measure of computational cost rather than a formal big $O$ notation. Thus, we refrained from a more tedious analysis of the computational cost in a formal manner. However, we are happy to add a brief discussion on computational complexity to the paper. However, we would like to clarify why we take this stance.
>
> Formally, the convolutional neural networks will have a constant resolution, thus a specified cost associated with the $64\times 64$ latent space diffusion and $256\times 256$ input and output of the auto-encoder. In big $O$ notation, this would be seen as a constant since we run the denoising with a fixed number of steps for all cases, thus $O(1)$. The neural field after the convolutional decoder simply scales by the number of pixels. In TO/FEA terms, this would be element count, thus it becomes overall $O(N)$, with $N$ being the number of elements or pixels. This formal analysis, however, does not to cover the reality that all of the deep learning aspects, which boil down to $O(N)$, can be run in a highly parallelized pipeline on the GPU. For the optimization problem, one has to solve a linear system of equations for each iteration of optimization, which naively is $O(N^3)$ for a genearal $N \times N$ matrix (Note in FEA the size of the system is actually $\text{Nodes} \times \text{dof}$ which becomes $O(N)$ when constants are taken care of). But at this point many experts in the numerical linear algebra field would point out that the problems we solve in FEA are structured and highly sparse, and thus can be solved with a multi-grid solver, which in theory reduces the linear system solve cost to a best-case scenario of $O(N)$[1-3]. But this again does not cover the fact that multi-grid solvers are preconditioned conjugate gradient solvers which operate by solving the system iteratively, thus making them not parallelizable, and therefore much slower. Which brings us back to the wall-clock being a much more reasonable measure of computational efficiency in this specific case. Regardless, we acknowledge that this nuance is not immediately clear. Given this, we will add the following brief discussion to the the paper, mentioning this high-level analysis and the reasoning for inference time being a better overall metric. Specifically, we propose the following change to line 378:
>
> > ...over iterative optimization. It's important to mention that a high-level analysis of computational cost in this case would result in an $O(N)$ complexity for OAT, where $N$ is the number of pixels/elements for a given sample. This is because the non-constant cost of OAT is in the neural field renderer, which would have a complexity of $O(N)$. This is compared to the expensive FEA simulations at each step of the optimization, yielding an $O(N^3)$ complexity (Solving a linear system of equations). However, given that FEA matrices in this setting are highly sparse and the meshes in our work are structured, specialized multi-grid solvers can, in theory, reach an $O(N)$ efficiency [1-3]. This high-level analysis is however, fails to realize the inference efficiency of deep learning models compared to conventional optimization, given that it will not take into account the highly parallelized inference of deep learning models on the GPU compared to the iterative nature of linear system solvers. This makes inference time an overall better measure of performance in practical consumer hardware.
>
> [1] Erik A. Träff, et. al. Simple and efficient GPU-accelerated topology optimisation: Codes and applications. Computer Methods in Applied Mechanics and Engineering
> [2]Haixiang Liu, et. al. Narrow-band topology optimization on a sparsely populated grid. ACM Trans. Graph.
> [3]Niels Aage, et. al. Giga-voxel computational morphogenesis for structural design. Nature.
>
> ## On Median Reporting Of Volume Fraction Error
> The main reason we omit this is that we did not explicitly discuss this aspect of the results, given that no further insight was gained from this, and the overall lower median was covered in the CE discussion. Most notably, removing these values helped make the tables more readable and helped report results in a more concise way. The code we provide includes the full benchmarking, which reports the medians of VFE. If you are curious about the values themselves, below are the full results of OAT in each table:
>
> **Table 1**
> | Model | CE % | CE % Med | VFE % | VFE % Med |
> | :--- | :--- | :--- | :--- | :--- |
> | OAT (Ours) | 0.169 | -0.016 | 2.21 | 2.03 |
> | OAT (Ours) + 5 | 0.0438 | 0.001 | 0.91 | 0.78 |
> | OAT (Ours) + 10 | 0.0329 | 0.0142 | 0.29 | 0.21 |
>
> **Table 2**
> | Model | CE % | CE % Med | VFE % | VFE % Med |
> | :--- | :--- | :--- | :--- | :--- |
> | OAT (Ours) | 0.89 | 0.23 | 1.42 | 1.22 |
> | OAT (Ours) + 5 | 0.34 | 0.11 | 0.53 | 0.38 |
> | OAT (Ours) + 10 | 0.042 | 0.016 | 0.089 | 0.076 |
>
> **Table 3**
> | Model | CE % | CE % Med | VFE % | VFE % Med |
> | :--- | :--- | :--- | :--- | :--- |
> | OAT (Ours) | 0.42 | 0.046 | 1.90 | 1.66 |
> | OAT (Ours) + 5 | 0.19 | 0.053 | 0.73 | 0.51 |
> | OAT (Ours) + 10 | 0.079 | 0.056 | 0.21 | 0.098 |
>
> **Table 4*
> | Model | CE* % | CE* % Med | VFE* % | VFE* % Med | Failure Rate % |
> | :--- | :--- | :--- | :--- | :--- | :--- |
> | OAT (Ours) | 10.06 | 4.49 | 1.03 | 0.97 | 48.55 |
> | OAT (Ours) + 5 | 7.37 | 3.37 | 0.70 | 0.65 | 21.46 |
> | OAT (Ours) + 10 | 5.69 | 2.32 | 0.67 | 0.59 | 15.51 |
>
> We hope that this clarifies the reasoning for omitting these values. If the reviewer believes this adds additional insight, we are happy to add this to the camera-ready paper.

---

> > ### Author Response · Authors · 2025-08-08
> >
> > We thank the reviewer for their thoughtful comments and appreciate the fact that the insight provided has helped improve the paper. We hope that the provided response and clarifications have been satisfactory to the reviewer.

---

### Decision · Program_Chairs · 2025-09-17

**Decision:**

Accept (poster)

**Comment:**

The paper introduces a foundation modeling approach to topological optimization, plus a large new data set of optimized solutions across different boundary conditions.

Both the problem, and the question of foundational approaches in this area, are of clear and timely interest to the machine learning community. All reviewers agree that there are some interesting contributions here, although opinions are somewhat divided regarding significance. Personally, I do share reviewer zRq4's concern regarding overclaiming: On the one hand, there are contributions and results here, on the other hand, the claims made about these results seem rather strong. All in all, however, the majority of reviewers believe those contributions warrant publication.